# Effective population size does not explain long-term variation in genome size and transposable element content in animals

Alba Marino[1]*, Gautier Debaecker[2], Anna-Sophie Fiston-Lavier[1,3], Annabelle Haudry[4], Benoit Nabholz[1,3]*

[1]ISEM, Université de Montpellier, CNRS, IRD, Montpellier, France; [2]Université Claude Bernard Lyon 1, LEHNA UMR 5023, CNRS, Villeurbanne, France; [3]Institut Universitaire de France, Paris, France; [4]Université Claude Bernard Lyon 1, LBBE UMR 5558, Villeurbanne, France

## eLife Assessment

This **important** study offers a powerful empirical test of a highly influential hypothesis in population genetics. It incorporates a large number of animal genomes spanning a broad phylogenetic spectrum and treats them in a rigorous unified pipeline, providing the **convincing** negative result that effective population size scales neither with the content of transposable elements nor with overall genome size. These observations demonstrate that there is still no simple, global hypothesis that can explain the observed variation in transposable element content and genome size in animals.

*For correspondence:
alba.marino@umontpellier.fr (AM);
benoit.nabholz@umontpellier.fr (BN)

Competing interest: The authors declare that no competing interests exist.

**Abstract** Animal genomes exhibit a remarkable variation in size, but the evolutionary forces responsible for such variation are still debated. As the effective population size ($Ne_e$) reflects the intensity of genetic drift, it is expected to be a key determinant of the fixation rate of nearly-neutral mutations. Accordingly, the Mutational Hazard Hypothesis postulates lineages with low $Ne_e$ to have bigger genome sizes due to the accumulation of slightly deleterious transposable elements (TEs), and those with high $Ne_e$ to maintain streamlined genomes as a consequence of a more effective selection against TEs. However, the existence of both empirical confirmation and refutation using different methods and different scales precludes its general validation. Using high-quality public data, we estimated genome size, TE content, and rate of non-synonymous to synonymous substitutions (dN/dS) as $Ne_e$ proxy for 807 species including vertebrates, molluscs, and insects. After collecting available life-history traits, we tested the associations among population size proxies, TE content, and genome size, while accounting for phylogenetic non-independence. Our results confirm TEs as major drivers of genome size variation, and endorse life-history traits and dN/dS as reliable proxies for $Ne_e$. However, we do not find any evidence for increased drift to result in an accumulation of TEs across animals. Within more closely related clades, only a few isolated and weak associations emerge in fishes and birds. Our results outline a scenario where TE dynamics vary according to lineage-specific patterns, lending no support for genetic drift as the predominant force driving long-term genome size evolution in animals.

## Introduction

The variation in genome size among animals is remarkable, spanning four orders of magnitude, from 0.02 Gb in the nematode *Pratylenchus coffeae* to 120 Gb in the marbled lungfish *Protopterus aethiopicus* (*Gregory, 2023*). Understanding why such a huge variation occurs is a long-standing

question in evolutionary biology. It is now established that genome size is not related to organismal complexity (C-value enigma) or the number of coding genes in eukaryotes. Rather, variation in DNA content depends on the amount of non-coding DNA such as transposable elements (TEs), introns, and pseudogenes (*Elliott and Gregory, 2015*; *Kidwell, 2002*; *Lynch et al., 2011*). However, the evolutionary mechanisms leading certain lineages to inflate their genome size or to maintain streamlined genomes are still debated (*Galtier, 2024*).

The various hypotheses that have been proposed can be divided into adaptive and non-adaptive. Adaptive theories such as the nucleoskeletal (*Cavalier-Smith, 1978*) and the nucleotypic one (*Gregory, 2001*; *Gregory and Hebert, 1999*) consider genome size to be mainly composed of 'indifferent' DNA (*Graur et al., 2015*), whose bulk is indirectly selected as a consequence of its effect on nuclear and cellular volume. Cellular phenotypes, in turn, are thought to influence the fitness of organisms by affecting traits such as cell division rate (*Bennett and Riley, 1997*), metabolic rate (*Vinogradov, 1995*; *Vinogradov, 1997*), and developmental time and complexity (*Jockusch, 1997*; *Olmo et al., 1989*; *Gregory, 2002*). On the other hand, non-adaptive theories emphasize the importance of the neutral processes of mutation and genetic drift in determining genome size (*Lynch and Conery, 2003*; *Petrov, 2001*). In particular, the concepts originally proposed by Lynch and Conery were later on formalized within the framework of the Mutational Hazard Hypothesis (MHH; *Lynch, 2007*).

The MHH posits that tolerance to the accumulation of non-coding DNA depends on its mutational liability, which is minimized when the mutation rate is low and the effective population size is high ($Ne_e$, which is inversely proportional to the intensity of genetic drift). The fundamental assumption is that most of such extra DNA is mildly deleterious and its fate in the population depends on the interplay between selection and genetic drift: a newly emerged nearly-neutral allele with a given negative selective effect ($s$) should be effectively removed from the population when $Ne_e$ is high ($|Ne_e s| \gg 1$), while it should have approximately the same chance of fixation of a neutral allele when $Ne_e$ is low ($|Ne_e s| \sim 1$ or lower than 1) as drift will be the predominant force (*Ohta, 1992*).

Because of the pervasiveness of TEs and their generally neutral or slightly deleterious effect (*Arkhipova, 2018*), their dynamics in response to changing $Ne_e$ are of particular interest in the context of the MHH. TE insertions are expected to drift to fixation as neutral alleles and enlarge genome size in organisms with low $Ne_e$, while the genomes of organisms with large $Ne_e$ are expected to remain streamlined as emerging TEs should be efficiently removed by purifying selection (*Lynch, 2007*). The MHH is highly popular: indeed, it is based on general principles of population genetics and proposes a unifying explanation for the evolution of complex traits of genome architecture without recourse to specific molecular processes. The studies supporting the MHH are based on phylogenetically very diverse datasets as the goal of the theory is to explain broad patterns of complexity emergence and variation (*Lynch and Conery, 2003*; *Yi and Streelman, 2005*; *Yi, 2006*). Nevertheless, other authors pointed out that the application of the MHH to such distantly related taxa could suffer from confounding factors intervening across organisms with very different biologies (*Charlesworth and Barton, 2004*; *Daubin and Moran, 2004*), thus raising the question whether $Ne_e$ could explain genome size variation patterns at finer scales. Additionally, potential issues of robustness of the original dataset to phylogenetic control have been raised (*Lynch, 2011*; *Whitney et al., 2011*; *Whitney and Garland, 2010*). On top of that, an alternative TE-host-oriented perspective is that the accumulation of TEs in particular depends on their type of activity and dynamics, as well as on the lineage-specific silencing mechanisms evolved by host genomes (*Ågren and Wright, 2011*).

Recent attempts have been made to assess the impact of increased genetic drift on the genomic TE content and genome size increase across closely related species with similar biological characteristics, employing both genetic diversity data and life-history traits (LHTs) as predictors of $Ne_e$. While some studies do not find any evidence supporting the role of $Ne_e$ in genome size and TE content variation (*Bast et al., 2016*; *Kapheim et al., 2015*; *Mackintosh et al., 2019*; *Roddy et al., 2021*; *Yang et al., 2024*), others do (*Chak et al., 2021*; *Cui et al., 2019*; *Lefébure et al., 2017*; *Mérel et al., 2021*; *Mérel et al., 2025*). Thus, a univocal conclusion can hardly be drawn from such studies. The MHH has thus been investigated at either very wide – from prokaryotes to multicellular eukaryotes – or narrow phylogenetic scales (i.e. inter-genera or inter-population). However, no study spanning across an exhaustive set of distantly related taxa and relying on a phylogenetic framework has yet been performed to our knowledge. Such an approach would allow a systematic test of the association between $Ne_e$ variation and long-term patterns of genome and TE expansion at a wider

evolutionary scale, while controlling for the effect of phylogenetic inertia. Synonymous genetic diversity is commonly used to inform patterns of $Ne_e$ (*Lynch and Conery, 2003*; *Romiguier et al., 2014*; *Buffalo, 2021*; *Lynch et al., 2023*). However, while its insights are limited to the age of current alleles (mostly less than $10Ne_e$ generations in diploid organisms), complex genomic features likely have much deeper origins. Comparative measures of divergence, like the genome-wide ratio of non-synonymous substitution rate to synonymous substitution rate (dN/dS), can quantify the level of genetic drift acting on protein-coding sequences since the last common ancestor. This approach accounts for processes occurring over a longer time scale than those responsible for genetic diversity. In fact, polymorphism might reflect relatively recent population size fluctuations (*Daubin and Moran, 2004*) and might even diverge from indices of long-term $Ne_e$ if the selection-drift equilibrium is not reestablished (*Lefébure et al., 2017*; *Müller et al., 2022*). We therefore adopt dN/dS as an index of long-term $Ne_e$, as it is more likely to reflect the evolutionary lapse during which the deep changes in genome size and TE content that we are investigating occurred (*Whitney et al., 2011*; *Whitney and Garland, 2010*).

In this study, we took advantage of 3214 public metazoan reference genomes and C-value records (i.e. the haploid DNA content of a nucleus) to estimate genome sizes. A subset of 807 species including birds, mammals, ray-finned fishes, insects and molluscs was selected to test the predictions of the MHH, especially through the relationship among the level of drift, genome size and genomic TE content. A phylogeny was computed with metazoan-conserved genes. $Ne_e$ was accounted for by the dN/dS and by LHTs when available; TE contents were estimated de novo from read data. Controlling for phylogenetic non-independence, we (1) assessed the contribution of TE quantity to genome size differences and (2) evaluated the efficacy of $Ne_e$ proxies in explaining genome size and TE content variation, across the whole dataset and within specific clades.

## Results

### Selection of high-quality assemblies

The reference genomes of the 3,214 metazoan species were downloaded via the NCBI genome database (*Supplementary file 1*). From these, the genomes with contig N50 ≥50 kb and available C-value record were employed for genome size estimation (see Results: Genome size estimation, Methods: Genome size estimation). Based on the assembly quality (contig N50 ≥50 kb), the completeness of metazoan core genes (complete BUSCO orthologs ≥70%) (*Manni et al., 2021*) and the raw sequencing data availability, a dataset of 930 genomes was then retained for downstream analyses. For reliable estimation of substitution rates, this dataset was further downsized to 807 representative genomes as species-poor, deep-branching taxa were excluded (*Figure 1*; *Table 3—source data 1*).

### Genome size estimation

For the selected genomes, the genome size records available for 465 species (*Figure 2—source data 1*) show that the assembly size is strongly positively correlated with the C-value (Pearson's $r$=0.97, p-value <0.001). This indicates the reliability of the use of assemblies to estimate genome size. Although a non-linear model is not statistically better than the Weighted Least Squares (WLS), assembly sizes tend to underestimate genome size in comparison to C-values, an effect becoming more and more evident as genome size increases (*Figure 2*). According to the metadata that we could retrieve, whether a genome was assembled using long read or uniquely short read data does not affect the slope of the WLS with assembly size as independent variable (T-test: long reads - p-value = 0.88; short reads - p-value = 0.87). Because it is not affected by sequencing biases, C-value was used when available; otherwise, the predicted C-value according to WLS was employed. The value chosen as genome size estimation is reported for each species in *Supplementary file 1*, *Table 2—source data 1*.

### Transposable elements and genome size variation

Repeat content of a subset of 29 dipteran genomes was previously estimated using EarlGrey v1.3 (*Baril et al., 2024*) and a wrapper around dnaPipeTE (*Goubert et al., 2015*), an assembly-based and a read-based pipeline, respectively. dnaPipeTE leverages the sampling of reads at low-coverage to perform de novo assembly of TE consensus sequences: this approach has the advantage of being unbiased by repeat sequences potentially missing from genome assemblies. The results of the two

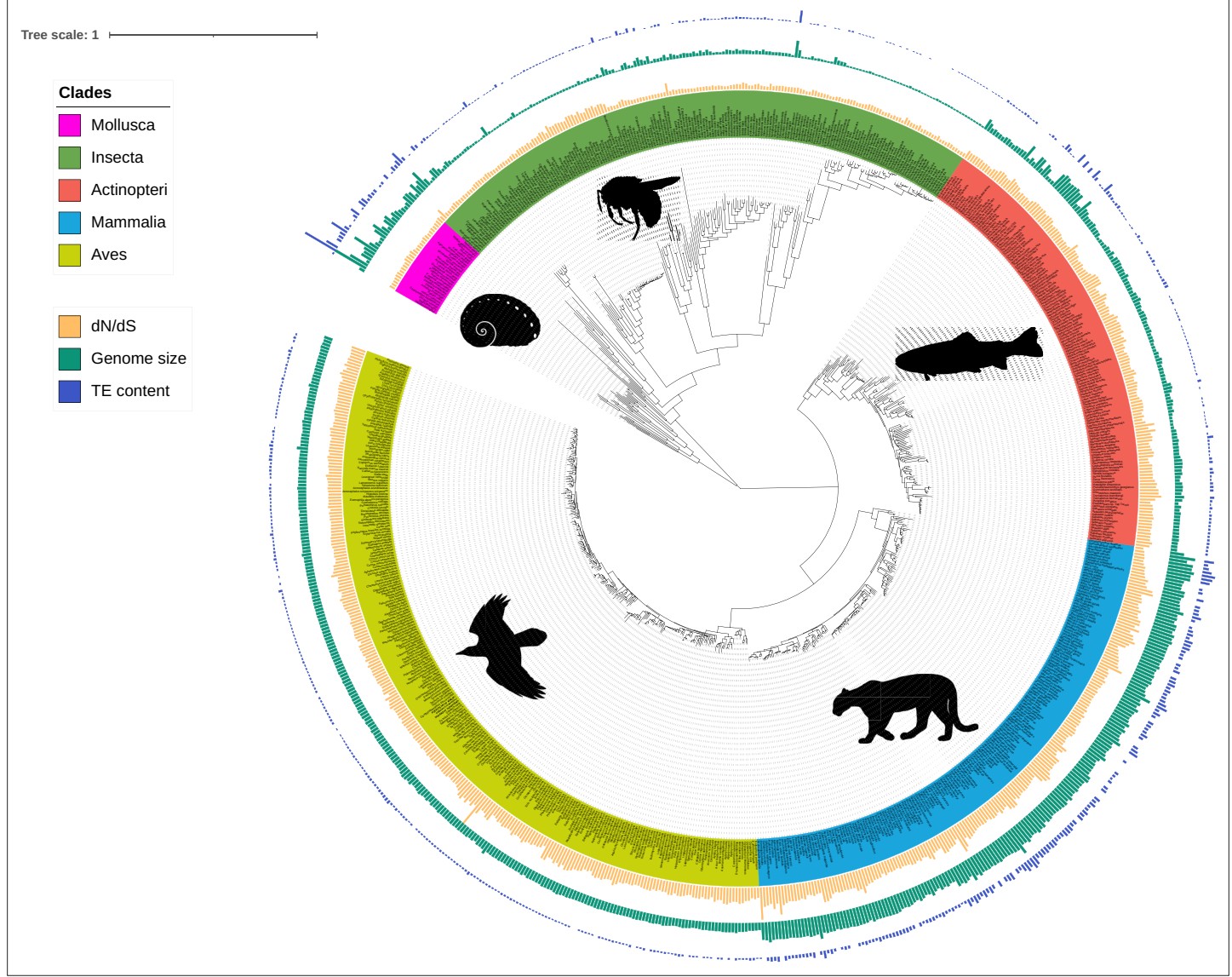

**Figure 1.** Phylogeny of the 807 species including ray-finned fishes (Actinopteri), birds (Aves), insects (Insecta), mammals (Mammalia), and molluscs (Mollusca). Bars correspond to TE content (bp, blue), genome size (bp, green), and dN/dS estimations (values between 0 and 1, yellow). Branch lengths are amino-acid substitutions calculated on BUSCO genes. The tree was plotted with iTOL (*Letunic and Bork, 2021*).

The online version of this article includes the following source data for figure 1:

**Source data 1.** iTOL annotation file for dN/dS bars.

**Source data 2.** iTOL annotation file for genome size bars.

**Source data 3.** iTOL annotation file for TE content bars.

methods overall agree with each other across the scanned genomes (Genomic percentage of TEs: Pearson's $r$=0.88, p-value <0.001; TE base pairs: Pearson's $r$=0.90, p-value <0.001), with the most notable difference being the proportion of unknown elements, generally higher in dnaPipeTE estimations (Wilcoxon signed-rank test, p-value <0.001; *Figure 3*; *Figure 3—figure supplement 1*; *Figure 3—source data 1*). We therefore mined the remaining genomes with dnaPipeTE which is much less computationally intensive. Repeat content could only be estimated for 672 species over the 807 representative genomes (*Table 3—source data 1*): for the remaining 135, the pipeline could not be run because of unsuitable reads (e.g. only long reads available or too low coverage).

A very strong positive correlation between TE content and genome size is found both across the whole dataset and within taxa (*Figure 4A*; *Figure 4—figure supplement 1*; *Table 1*, *Table 2*). A

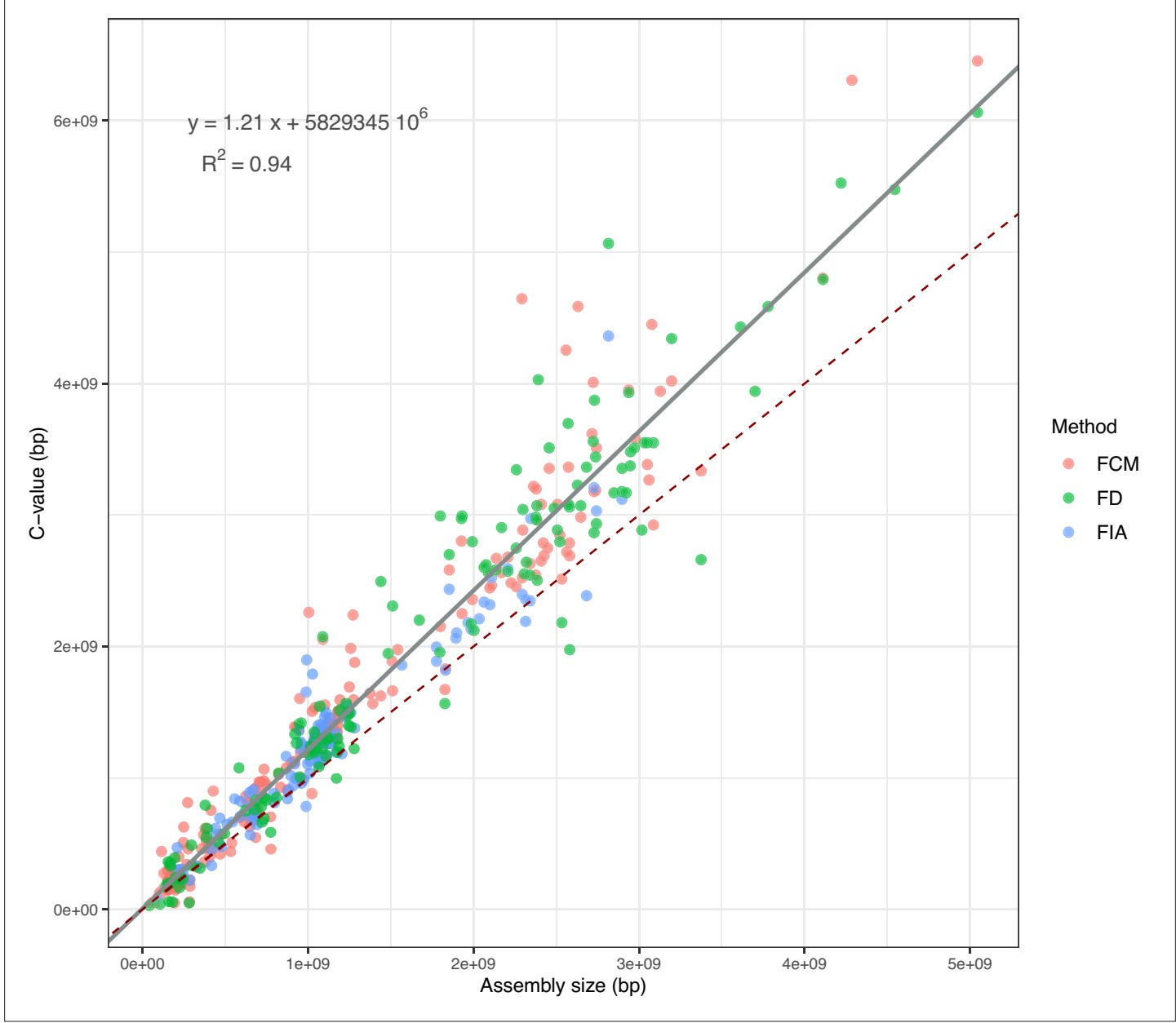

**Figure 2.** Correlation between assembly sizes and C-values for 365 species with contig N50 ≥50 kb. The grey slope corresponds to the WLS used to predict the expected C-values (reported in the equation). The dark-red dashed slope marks the hypothetical 1:1 relationship. FCM = Flow Cytometry, FD = Feulgen Densitometry, FIA = Feulgen Image Analysis.

The online version of this article includes the following source data for figure 2:

**Source data 1.** C-value records and assembly sizes used to train the WLS.

notable exception concerns the avian clade that deviates from this pattern: the range of TE content is wider than the one of genome size compared to the other clades (*Figure 4A*), resulting in a weaker power of TEs in explaining genome size variation in this group (*Tables 1 and 2*).

Alternatively to the impact of TEs on genome size, we investigated whether whole or partial genome duplications could be major factors in genome size variation among animals. BUSCO Duplicated score has indeed a slightly positive correlation with genome size, which is however much weaker than that of TEs (Slope = $6.639 \cdot 10^{-9}$, adjusted-$R^2$=0.022, p-value <0.001). Of the 24 species with more than 30% of duplicated BUSCO genes, 13 include sturgeon, salmonids, and cyprinids, known to have undergone whole genome duplication (*Du et al., 2020*; *Li and Guo, 2020*; *Lien et al., 2016*), and five

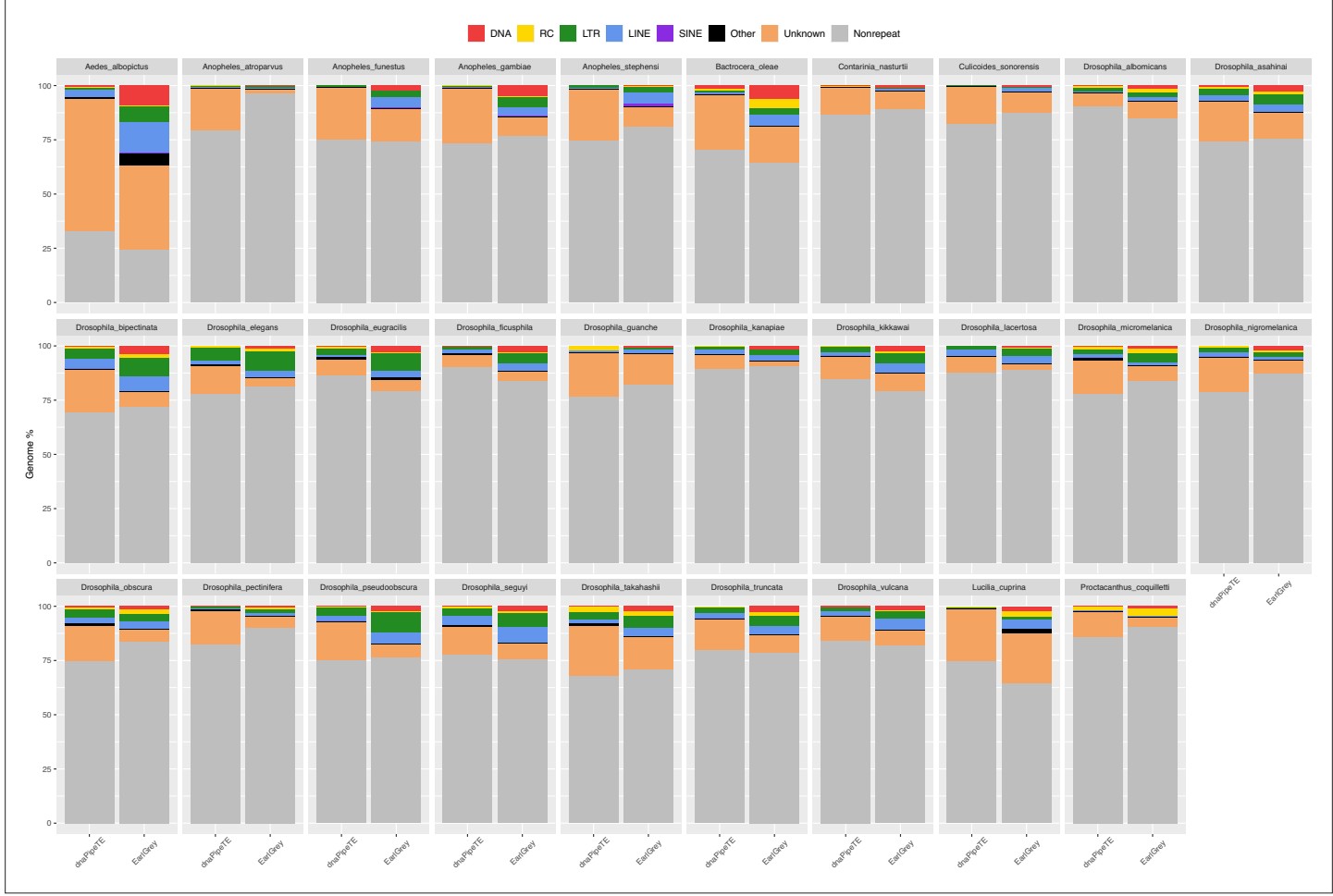

**Figure 3.** Genomic proportion occupied by repeats in 29 dipteran species as estimated by EarlGrey and by the dnaPipeTE wrapper pipelines. The genome percentage is calculated proportionally to the assembly size in the case of EarlGrey, while it is calculated in relation to the genome size estimated in this study in the case of dnaPipeTE. DNA = DNA elements; RC = Rolling Circle; LTR = Long Terminal Repeats; LINE = Long Interspersed Nuclear Elements; SINE = Short Interspersed Nuclear Elements. 'Other' includes simple repeats, microsatellites, RNAs. 'Unknown' includes all repeats that could not be classified.

The online version of this article includes the following source data and figure supplement(s) for figure 3:

**Source data 1.** Quantity of repeated elements for 29 dipteran genomes as estimated by EarlGrey and dnaPipeTE.

**Figure supplement 1.** Genomic length occupied by repeats in 29 dipteran species as estimated by EarlGrey and by the dnaPipeTE wrapper pipelines.

are dipteran species, where gene duplications are common (*Ruzzante et al., 2019*). In general, TEs appear as the main factor influencing genome size variation across species.

## dN/dS and life history traits as proxies of effective population size

Intensity of effective selection acting on species can be informed by the dN/dS ratio: a dN/dS closer to 1 accounts for more frequent accumulation of mildly deleterious mutations over time due to increased genetic drift, while a dN/dS close to zero is associated with a stronger effect of purifying selection. We therefore employed this parameter as a genomic indicator of $Ne_e$, as the two are expected to scale negatively between each other. We compiled several LHTs from different sources (see Methods: Compilation of life history traits) to cross-check our estimations of dN/dS. In general, dN/dS is expected to scale positively with body length, age at first birth, maximum longevity, age at sexual maturity and mass, and to scale negatively with metabolic rate, population density and depth range.

We estimated dN/dS with a mapping method (hereafter referred to as Bio ++ dN/dS; *Dutheil et al., 2006*; *Guéguen et al., 2013*), and with a bayesian approach using Coevol (hereafter referred to as Coevol dN/dS; *Lartillot and Poujol, 2011*). The two metrics are reported in *Table 3—source data*

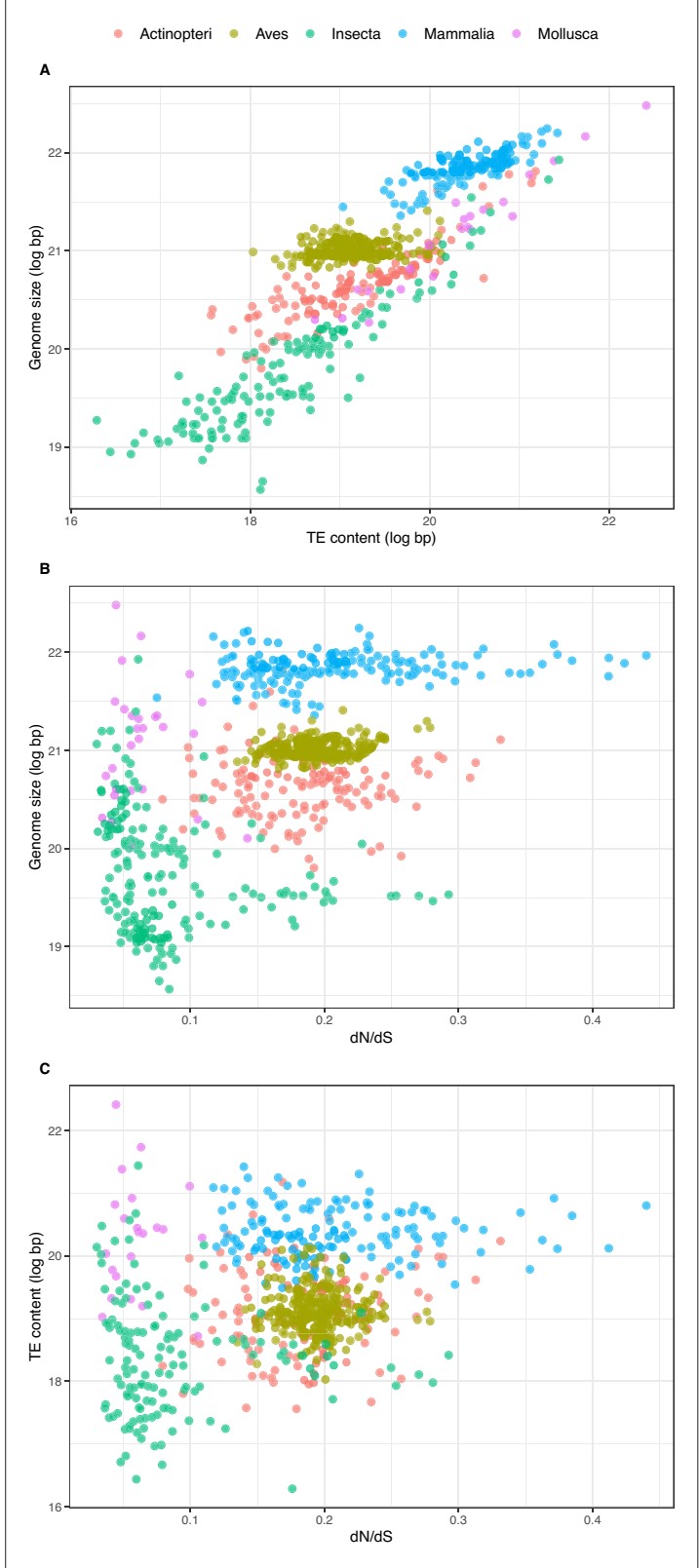

**Figure 4.** Relationship between overall TE content, genome size, and dN/dS. (**A**) Relationship between overall TE content and genome size (N=672, log-transformed): slope = 0.718, adjusted-$R^2$=0.751, p-value <0.001. (**B**) Relationship between genome size and dN/dS (N=785): slope = 6.100, adjusted-$R^2$=0.275, p-value <0.001. (**C**) Relationship between TE content and dN/dS (N=672): slope = 4.253, adjusted-$R^2$=0.092, p-value <0.001.

*Figure 4 continued on next page*

*Figure 4 continued*

Statistics refer to linear regression, see figure supplements and **Tables 1 and 3** for Phylogenetic Independent Contrasts results.

The online version of this article includes the following figure supplement(s) for figure 4:

**Figure supplement 1.** PIC regression of overall TE content as a predictor of genome size across the full dataset.

**Figure supplement 2.** PIC regressions of Coevol dN/dS estimated from the GC3-poor geneset as predictor of genome size, overall and recent TE content.

**Figure supplement 3.** PIC regressions for Coevol dN/dS estimated from the GC3-poor geneset as predictor of body mass and longevity.

*1.* Note that for Coevol, we report both the results relative to dN/dS at terminal branches (**Table 3**) and the correlations inferred by the model (**Tables 2 and 4**).

As expected, Bio ++ dN/dS scales positively with body mass and longevity under Phylogenetic Independent Contrasts (PIC) (**Table 3**; **Figure 4—figure supplement 3**). dN/dS estimation on the trimmed phylogeny deprived of short and long branches results in a stronger correlation with LHTs, suggesting that short branches might contribute to dN/dS fluctuations (**Table 3**; **Figure 5**). Coevol dN/dS values are however highly concordant with Bio ++ dN/dS (**Figure 6**) and scale positively with body mass and longevity, as well (**Table 3**).

As for Coevol reconstruction, dN/dS covaries as expected with most of the tested LHTs: dN/dS scales positively with body length, longevity, mass, sexual maturity, and depth range in fishes (**Table 2**); the same is found in mammals, in addition to negative correlations with population density and metabolic rate; in birds, mass, metabolic rate, and sexual maturity correlate in the same way with dN/dS, although this is consistently observed only for the GC3-rich gene set (**Table 4**). Based on the available traits, the estimations of dN/dS ratios obtained using two different methods correspond in general to each other, supporting dN/dS as a meaningful indicator of long-term effective population size, at least for vertebrate clades. Results are not reported for molluscs and insects, as none and very few records of LHTs (seven species with at least one trait) were available, respectively.

## dN/dS does not predict genome size and overall TE content across metazoans

If increased genetic drift leads to TE expansions, a positive relationship between dN/dS and TE content, and more broadly with genome size, should be observed. However, we find no statistical support for this relationship across all species in the PIC analysis. Similarly, no association is found when short and long branches are removed (**Figure 4B and C**; **Figure 4—figure supplement 2**; **Table 3**). Contrary to our expectations, Coevol dN/dS scales negatively with genome size across the whole dataset (Slope = –0.287, adjusted-$R^2$=0.004, p-value = 0.039) and within insects (Slope = –1.241, adjusted-$R^2$=0.026, p-value = 0.018).

Surprisingly, different patterns are observed relative to TE content, as a negative correlation with Coevol dN/dS is detected across all species (Slope = –0.903, adjusted-$R^2$=0.004, p-value = 0.050) and

**Table 1.** Correlation between genome size and overall TE content based on phylogenetic independent contrasts.

Statistics are shown relative to the overall dataset and to each clade. Variables were log-transformed previous to regression. Original values used to infer PIC statistics are included in *Table 3—source data 1*. * 0.05<p ≤ 0.01; ** 0.01<p ≤ 0.001; *** p<0.001. Significant correlations are highlighted in bold.

| | Regression coefficient | Adjusted-$R^2$ | p-value |
|---|---|---|---|
| Overall Dataset | 0.219 *** | 0.417 *** | <0.001 |
| Actinopteri | 0.300 *** | 0.610 *** | <0.001 |
| Aves | 0.042 *** | 0.039 ** | 0.001 |
| Insecta | 0.356 *** | 0.626 *** | <0.001 |
| Mammalia | 0.200 *** | 0.526 *** | <0.001 |
| Mollusca | 0.605 *** | 0.895 *** | <0.001 |

**Table 2.** Coevol correlations between genomic traits – genome size, TE content, and recent TE content – and LHTs. Different LHTs are shown according to availability for a clade. Posterior probabilities lower than 0.1 indicate significant negative correlations; posterior probabilities higher than 0.9 indicate significant positive correlations. Expected, significant correlations are marked in bold black; significant correlations opposite to the expected trend are marked in bold red.

| Coevol correlations | Actinopteri GC3-poor CC | PP | GC3-rich CC | PP | Aves GC3-poor CC | PP | GC3-rich CC | PP | Insecta GC3-poor CC | PP | GC3-rich CC | PP | Mammalia GC3-poor CC | PP | GC3-rich CC | PP | Mollusca GC3-poor CC | PP | GC3-rich CC | PP |
|---|---|---|---|---|---|---|---|---|---|---|---|---|---|---|---|---|---|---|---|---|
| **Genome size** | CC | PP | CC | PP | CC | PP | CC | PP | CC | PP | CC | PP | CC | PP | CC | PP | CC | PP | CC | PP |
| Body length | 0.015 | 0.56 | 0.015 | 0.56 | | | | | | | | | –0.065 | 0.26 | –0.081 | 0.21 | | | | |
| Basal metabolic rate (ml/O2/hour)* | | | | | | | | | | | | | 0.028 | 0.60 | 0.064 | 0.72 | | | | |
| Age at first birth | | | | | | | | | | | | | –0.223 | 0.02 | –0.256 | 0.01 | | | | |
| Population density | | | | | | | | | | | | | 0.203 | 0.97 | 0.200 | 0.96 | | | | |
| Maximum longevity | 0.135 | 0.85 | 0.142 | 0.87 | **0.116** | **0.91** | **0.161** | **0.94** | | | | | –0.106 | 0.14 | –0.109 | 0.15 | | | | |
| Mass | 0.030 | 0.61 | 0.040 | 0.65 | **0.201** | **1** | **0.185** | **0.99** | | | | | –0.077 | 0.23 | –0.097 | 0.16 | | | | |
| Metabolic rate (W)† | | | | | **–0.274** | **0.003** | **–0.210** | **0.02** | | | | | 0.047 | 0.68 | 0.078 | 0.76 | | | | |
| Sexual maturity | –0.011 | 0.45 | 0.013 | 0.52 | 0.039 | 0.64 | 0.048 | 0.67 | | | | | –0.176 | 0.05 | –0.180 | 0.05 | | | | |
| Depth range | –0.013 | 0.47 | 0.000 | 0.51 | | | | | | | | | | | | | | | | |
| Overall TE content | **0.793** | **1** | **0.779** | **1** | **0.293** | **1** | **0.335** | **1.00** | **0.842** | **1** | **0.834** | **1** | **0.679** | **1.00** | **0.678** | **1.00** | **0.909** | **1** | **0.895** | **1** |
| Recent TE content | **0.664** | **1** | **0.632** | **1** | **0.237** | **1** | **0.283** | **1.00** | **0.809** | **1** | **0.804** | **1** | **0.484** | **1.00** | **0.511** | **1.00** | **0.899** | **1** | **0.897** | **1** |
| **TE content** | CC | PP | CC | PP | CC | PP | CC | PP | CC | PP | CC | PP | CC | PP | CC | PP | CC | PP | CC | PP |
| Body length | –0.103 | 0.17 | –0.073 | 0.26 | | | | | | | | | **–0.238** | **0.013** | **–0.274** | **0.01** | | | | |
| Basal metabolic rate (ml/O2/hour)* | | | | | | | | | | | | | **0.164** | **0.89** | **0.253** | **0.97** | | | | |
| Age at first birth | | | | | | | | | | | | | **–0.427** | **0** | **–0.473** | **0.00** | | | | |
| Population density | | | | | | | | | | | | | **0.274** | **0.98** | **0.271** | **0.98** | | | | |
| Maximum longevity | 0.137 | 0.85 | 0.155 | 0.87 | 0.077 | 0.79 | 0.03 | 0.62 | | | | | **–0.231** | **0.013** | **–0.277** | **0.02** | | | | |
| Mass | –0.091 | 0.22 | –0.051 | 0.32 | –0.022 | 0.38 | –0.012 | 0.43 | | | | | **–0.233** | **0.017** | **–0.278** | **0.01** | | | | |
| Metabolic rate (W)† | | | | | 0.164 | 0.94 | 0.168 | 0.92 | | | | | **0.197** | **0.94** | **0.269** | **0.98** | | | | |
| Sexual maturity | –0.031 | 0.43 | 0.011 | 0.53 | –0.100 | 0.2 | –0.13 | 0.11 | | | | | **–0.329** | **0** | **–0.343** | **0.00** | | | | |
| Depth range | –0.026 | 0.43 | 0.004 | 0.52 | | | | | | | | | | | | | | | | |
| **Recent TE content** | CC | PP | CC | PP | CC | PP | CC | PP | CC | PP | CC | PP | CC | PP | CC | PP | CC | PP | CC | PP |
| Body length | –0.133 | 0.12 | –0.085 | 0.22 | | | | | | | | | **–0.297** | **0.00** | **–0.305** | **0.01** | | | | |
| Basal metabolic rate (ml/O2/hour)* | | | | | | | | | | | | | **0.229** | **0.94** | **0.308** | **0.99** | | | | |
| Age at first birth | | | | | | | | | | | | | **–0.486** | **0.00** | **–0.506** | **0.00** | | | | |
| Population density | | | | | | | | | | | | | **0.238** | **0.94** | **0.227** | **0.95** | | | | |
| Maximum longevity | 0.138 | 0.85 | **0.176** | **0.9** | 0.063 | 0.73 | 0.020 | 0.57 | | | | | **–0.334** | **0.00** | **–0.347** | **0.01** | | | | |
| Mass | **–0.153** | **0.10** | –0.096 | 0.21 | –0.026 | 0.36 | –0.007 | 0.47 | | | | | **–0.286** | **0.00** | **–0.308** | **0.01** | | | | |
| Metabolic rate (W)† | | | | | **0.142** | **0.9** | 0.144 | 0.88 | | | | | **0.265** | **0.97** | **0.313** | **0.99** | | | | |
| Sexual maturity | –0.011 | 0.47 | 0.051 | 0.62 | –0.126 | 0.15 | **–0.151** | **0.1** | | | | | **–0.329** | **0.00** | **–0.326** | **0.01** | | | | |
| Depth range | –0.036 | 0.39 | 0.040 | 0.61 | | | | | | | | | | | | | | | | |

*PanTHERIA.
†AnAge.

The online version of this article includes the following source data for table 2:

**Source data 1.** Genome sizes, all LHTs records, overall, and recent TE contents for the selected 807 species.

within Mammalia (Slope = –2.113, adjusted-$R^2$=0.063, p-value = 0.001), while a positive correlation is found within Actinopteri (Slope = 3.407, adjusted-$R^2$=0.046, p-value = 0.013; *Table 3*). Therefore, the two only significant positive PIC correlations found for birds and fishes are contrasted by results with an opposite trend in other groups. However, such correlations are slightly significant and their explained variance is extremely low.

**Table 3.** PIC results for the correlations of LHTs, genome size, and TE content against dN/dS.

Results for Bio ++ dN/dS are shown for the full dataset and for the phylogeny deprived of the longest (>1 amino-acid substitutions) and shortest (<0.01 amino-acid substitutions) terminal branches. Results for Coevol dN/dS are relative to the GC3-poor geneset. Only body mass and longevity are reported as LHTs (for an overview of all traits, see *Table 2*). For genomic traits, statistics are reported relative to the overall dataset and to each clade. Expected significant correlations of dN/dS with LHTs and genomic traits are marked in bold black; significant correlations opposite to the expected trend are marked in bold red. * 0.05 < p ≤ 0.01; ** 0.01 < p ≤ 0.001; *** p < 0.001.

| PIC correlations | | Bio++ (full phylogeny) | | | Bio++ (trimmed phylogeny) | | | Coevol | | |
|---|---|---|---|---|---|---|---|---|---|---|
| | | Regression coefficient | Adjusted-R² | p-value | Regression coefficient | Adjusted-R² | p-value | Regression coefficient | Adjusted-R² | p-value |
| Body mass (log gr)~dN/dS | Overall Dataset | 6.865 *** | 0.036 *** | <0.001 | 23.905 *** | 0.044 *** | <0.001 | 11.422 *** | 0.087 *** | <0.001 |
| Longevity (log years)~dN/dS | Overall Dataset | 2.147 ** | 0.025 ** | 0.004 | 11.349 *** | 0.097 *** | <0.001 | 2.970 *** | 0.050 *** | <0.001 |
| | Overall Dataset | 0.199 | 0.001 | 0.175 | 0.114 | –0.002 | 0.858 | –0.287 * | 0.004 * | 0.039 |
| | Actinopteri | 0.915 | 0.016 | 0.066 | 0.687 | –0.009 | 0.701 | 0.282 | –0.005 | 0.583 |
| | Aves | 0.109 | 0.001 | 0.270 | 0.709 | 0.013 | 0.073 | 0.238 | –0.001 | 0.407 |
| | Insecta | 1.085 | –0.002 | 0.411 | –3.703 | 0.005 | 0.204 | –1.241 * | 0.026 * | 0.018 |
| | Mammalia | 0.071 | –0.005 | 0.701 | 0.220 | –0.011 | 0.777 | –0.165 | 0.003 | 0.227 |
| Genome size (log bp)~dN/dS | Mollusca | 3.504 | –0.032 | 0.698 | 3.504 | –0.032 | 0.698 | –1.488 | –0.032 | 0.699 |
| | Overall Dataset | 0.798 | 0.004 | 0.062 | –0.216 | –0.002 | 0.899 | –0.903 * | 0.004 * | 0.050 |
| | Actinopteri | 2.139 | 0.015 | 0.098 | 1.393 | –0.012 | 0.751 | 3.407 * | 0.046 * | 0.013 |
| | Aves | 0.340 | –0.002 | 0.513 | –1.492 | –0.004 | 0.551 | 2.285 | 0.006 | 0.129 |
| | Insecta | 1.744 | –0.004 | 0.528 | –3.096 | –0.007 | 0.602 | 0.063 | –0.007 | 0.960 |
| | Mammalia | 1.001 | 0.0027 | 0.238 | 1.023 | –0.012 | 0.691 | –2.113 ** | 0.063 ** | 0.001 |
| TE content (log bp)~dN/dS | Mollusca | 22.930 | 0.026 | 0.225 | 22.930 | 0.026 | 0.225 | 0.881 | –0.050 | 0.936 |
| | Overall Dataset | 1.963 ** | 0.012 ** | 0.003 | 0.691 | –0.002 | 0.727 | –1.225 | 0.0029 | 0.089 |
| | Actinopteri | 2.271 | 0.013 | 0.113 | 1.241 | –0.012 | 0.793 | 4.365 ** | 0.061 ** | 0.005 |
| | Aves | 0.545 | –0.001 | 0.384 | –0.385 | –0.006 | 0.890 | 4.982 ** | 0.028 ** | 0.006 |
| | Insecta | 1.725 | –0.004 | 0.530 | –4.792 | –0.004 | 0.428 | 0.536 | –0.006 | 0.668 |
| | Mammalia | 4.115 * | 0.024 * | 0.031 | 3.192 | –0.006 | 0.460 | –3.151 * | 0.024 * | 0.032 |
| Recent TE content (log bp)~dN/dS | Mollusca | 14.730 | –0.024 | 0.481 | 14.730 | –0.024 | 0.481 | 2.986 | –0.047 | 0.8026 |

The online version of this article includes the following source data for table 3:

**Source data 1.** BUSCO Duplicated scores, genome sizes, body mass and longevity records, dN/dS estimations from Bio ++ and Coevol, overall and recent TE contents for the selected 807 species.

Overall, we find no evidence for a recursive association of long-term $Ne_e$ variation, as approximated by dN/dS, with genome size and TE content across the analysed animal taxa. PIC analysis without the 24 species with more than 30% of duplicated BUSCO genes produced similar results with dN/dS as independent variable (genome size: slope = 0.234, adjusted-$R^2$=0.002, p-value = 0.102; TE content: slope = 0.819, adjusted-$R^2$=0.004, p-value = 0.054; Recent TE content: slope = 2.002, adjusted-$R^2$=0.012, p-value = 0.003), indicating that genomic duplications have a negligible effect on the missing link between dN/dS and genome size.

## Population size and genome size: a complex relationship across clades

Although no strong signal is found across the full dataset using PIC, different trends within different clades are suggested by both PIC and Coevol approaches.

Coevol infers a negative correlation of dN/dS with genome size in insects (GC3-poor: CC = –0.330, p=0.03; GC3-rich: CC = –0.110, p=0.24) and TE content in mammals (GC3-poor: CC = –0.220, p=0.08; GC3-rich: CC = –0.249, p=0.06), and a positive correlation (even though below significance threshold) with TEs in fishes (GC3-poor: CC = 0.192, p=0.88; GC3-rich: CC = 0.167, p=0.84). Additionally, a negative correlation with TE content is found in birds for the GC3-rich geneset, while a positive – yet

**Table 4.** Correlation coefficients (CC) and posterior probabilities (PP) estimated by Coevol with the GC3-poor and GC3-rich genesets for the coevolution of dN/dS with life history and genomic traits.

Different LHTs are shown according to availability for a clade. Posterior probabilities lower than 0.1 indicate significant negative correlations; posterior probabilities higher than 0.9 indicate significant positive correlations. Expected significant correlations of dN/dS with LHTs and genomic traits are marked in bold black; significant correlations opposite to the expected trend are marked in bold red. The original LHTs values used as input for Coevol are the same as those reported in *Table 2—source data 1*.

| Coevol correlations | Actinopteri dN/dS | | | | Aves dN/dS | | | | Insecta dN/dS | | | | Mammalia dN/dS | | | | Mollusca dN/dS | | | |
|---|---|---|---|---|---|---|---|---|---|---|---|---|---|---|---|---|---|---|---|---|
| | GC3-poor | | GC3-rich | | GC3-poor | | GC3-rich | | GC3-poor | | GC3-rich | | GC3-poor | | GC3-rich | | GC3-poor | | GC3-rich | |
| | CC | PP | CC | PP | CC | PP | CC | PP | CC | PP | CC | PP | CC | PP | CC | PP | CC | PP | CC | PP |
| Body length (cm) | 0.246 | 0.92 | 0.289 | 0.97 | | | | | | | | | 0.371 | 1 | 0.271 | 0.99 | | | | |
| Basal metabolic rate (ml/O₂/hour)* | | | | | | | | | | | | | -0.258 | 0.02 | -0.337 | 0.01 | | | | |
| Age at first birth (days) | | | | | | | | | | | | | 0.428 | 1 | 0.384 | 1 | | | | |
| Population density (individuals/km²) | | | | | | | | | | | | | -0.361 | 0.001 | -0.188 | 0.09 | | | | |
| Maximum longevity (years) | 0.435 | 0.97 | 0.330 | 0.95 | 0.303 | 0.92 | 0.109 | 0.75 | 0.0984 | 0.62 | 0.157 | 0.67 | 0.295 | 0.99 | 0.376 | 1 | | | | |
| Mass (g) | 0.223 | 0.89 | 0.308 | 0.97 | 0.096 | 0.84 | 0.155 | 0.95 | 0.0243 | 0.52 | 0.221 | 0.76 | 0.412 | 1 | 0.316 | 1 | | | | |
| Metabolic rate (Watt)† | | | | | 0.030 | 0.57 | -0.244 | 0.07 | 0.201 | 0.78 | -0.038 | 0.45 | -0.283 | 0.02 | -0.377 | 0.003 | | | | |
| Sexual maturity (days) | 0.320 | 0.91 | 0.286 | 0.88 | 0.032 | 0.58 | 0.284 | 0.99 | -0.244 | 0.19 | 0.072 | 0.58 | 0.468 | 1 | 0.329 | 1 | | | | |
| Depth range | 0.388 | 0.93 | 0.584 | 1 | | | | | | | | | | | | | | | | |
| Genome size (bp) | 0.034 | 0.59 | -0.027 | 0.42 | 0.054 | 0.69 | 0.072 | 0.77 | -0.330 | 0.03 | -0.110 | 0.24 | -0.041 | 0.35 | -0.140 | 0.12 | 0.142 | 0.65 | 0.306 | 0.82 |
| Overall TE content (bp) | 0.192 | 0.88 | 0.167 | 0.84 | 0.065 | 0.73 | -0.195 | 0.03 | -0.203 | 0.16 | -0.017 | 0.45 | -0.220 | 0.08 | -0.249 | 0.06 | -0.119 | 0.39 | -0.021 | 0.47 |
| Recent TE content (bp) | 0.205 | 0.89 | 0.258 | 0.95 | 0.109 | 0.83 | -0.240 | 0.01 | -0.136 | 0.25 | 0.049 | 0.63 | -0.119 | 0.23 | -0.296 | 0.06 | -0.172 | 0.34 | -0.004 | 0.50 |

*PanTHERIA.

†AnAge.

not significant – trend is found using the GC3-poor geneset (GC3-poor: CC = 0.065, p=0.73; GC3-rich: CC = –0.195, p=0.03; *Table 4*). Even though available for fewer species, LHTs partially support these trends for vertebrates: Actinopteri display a positive correlation between longevity and recent TE content (see Results: dN/dS weakly correlates with the recent TE content). Instead, mammalian TE content correlates positively with metabolic rate and population density, and negatively with body length, mass, sexual maturity, age at first birth and longevity (*Table 2*). Within Aves, Coevol predicts opposite results for genome size and TE content: genome size associates positively with longevity and mass, as well as with dN/dS, and negatively with metabolic rate, while TEs correlate positively with metabolic rate (and negatively with dN/dS in one case; *Table 2*). In summary, $Ne_e$ seems to negatively affect TE content in fishes, and positively in mammals. Importantly, genome size correlations seem to follow the same trends of TE content in these groups, although correlations are weaker and mostly non-significant. In the case of birds, genome size seems rather to be explained by $Ne_e$ as expected by MHH but not TE content, which instead might have the opposite trend.

## dN/dS weakly correlates with the recent TE content

The global TE content integrally reflects a long history of TE insertions and deletions. To have a glance at the dynamics of TEs on an evolutionary time comparable to that of the level of drift estimated using dN/dS, we additionally examined the quantity of the youngest elements. From the overall TE insertions, we estimated a recent TE content, defined by reads with less than 5% of divergence from consensus, and included it among the traits to model with PIC and Coevol. In PIC analysis, the variation of recent TE content weakly associates with Bio ++ dN/dS across the full dataset (Slope = 1.963, adjusted-$R^2$ = 0.012, p-value = 0.003) and in mammals (Slope = 4.115, adjusted-$R^2$=0.024, p-value = 0.031). On the contrary, using Coevol dN/dS this correlation is found in Aves (Slope = 4.982, adjusted-$R^2$=0.028, p-value = 0.006) and Actinopteri (Slope = 4.365, adjusted-$R^2$=0.061, p-value = 0.005), but the opposite is detected in mammals (Slope = –3.151, adjusted-$R^2$=0.024, p-value = 0.032) (*Table 3*). In agreement with PIC, Coevol reconstruction retrieves a positive association of recent TE

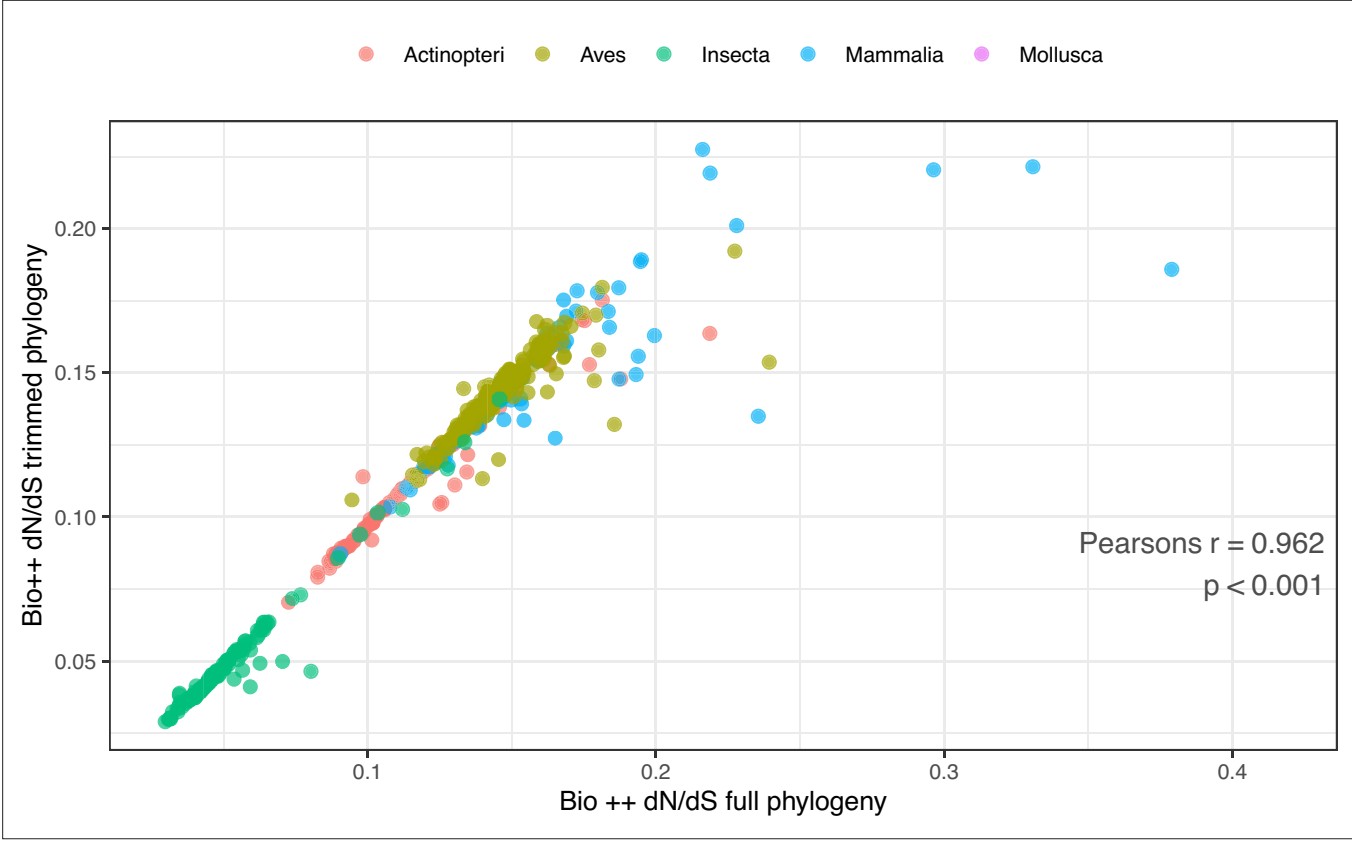

**Figure 5.** Comparison of Bio ++ dN/dS estimated from the full and pruned phylogenies. To obtain the pruned phylogeny, branches longer than 1 and shorter than 0.01 amino-acid substitutions were removed, leaving 485 tips. Pearson's *r*=0.962, p-value <0.001. The corresponding dN/dS values are included in *Table 3—source data 1*.

content with dN/dS and longevity only in fishes, and a relationship opposite to expectations with dN/dS (CC = –0.296, p=0.06) and LHTs in mammals. In contrast with PIC results, a negative correlation between recent TE content and dN/dS is found for birds using the GC3-rich genesets (CC = –0.240, p=0.01; *Tables 2 and 4*).

On the whole, only a very weak positive correlation of dN/dS with recent TE insertions is observed across all species. However, considering again the taxa separately, clade-specific patterns emerge: a negative association between population size proxies and recent TE content is jointly found by the two methods only in fishes. Conversely, mammals show a positive correlation between recent TE content and population size proxies. Therefore, the coevolution patterns between population size and recent TE content are consistent with the pictures emerging from the comparison of population size proxies with genome size and overall TE content in the corresponding clades.

## Discussion

Our results demonstrate the absence of a negative relationship between genome size and effective population size across a large dataset of animals, in contrast to the prediction of the MHH (*Lynch and Conery, 2003*; *Lynch, 2007*). Rather, our results highlight heterogeneous patterns within clades, with no consistent response of genome size and TE dynamics to $Ne_e$ variations.

### Assembly size underestimate genome size as genomes grow bigger

Assembly size is commonly used as a measure of genome size. However, the difficulty in assembling repetitive regions generally have it underestimate the actual genome size, in particular when only short reads are employed (*Benham et al., 2024*; *Blommaert, 2020*; *Peona et al., 2018*). Consequently, methods that directly measure C-value such as flow cytometry (*Dolezel and Greilhuber,*

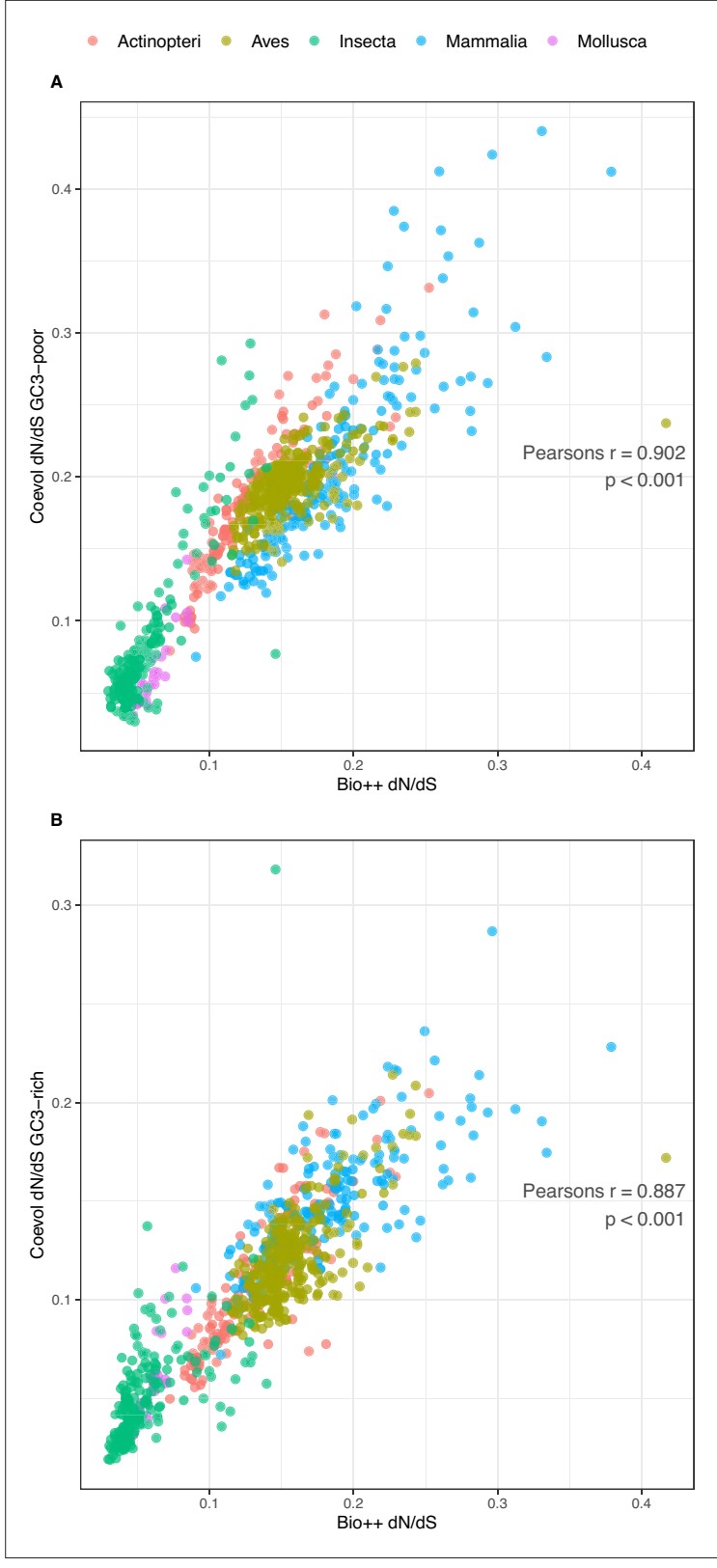

**Figure 6.** Comparison of Bio ++ and Coevol dN/dS estimations. (**A**) GC3-poor geneset (N=785): Pearson's *r*=0.902, p-value <0.001. (**B**) GC3-rich geneset (N=785): Pearson's *r*=0.887, p-value <0.001. The corresponding dN/dS values are included in **Table 3—source data 1**.

*2010*) and Feulgen densitometry (*Hardie et al., 2002*) are normally preferred as they do not rely on sequence data. Despite applying quality criteria to the assembly, the relationship between assembly size and genome size might still be questioned. However, we show that assembly size can overall approximate genome size quite well and, probably because we removed lower quality assemblies, no effect related to read type (short Illumina *vs* long ONT/PacBio) was detected. This suggests that the assemblies selected for our dataset can mostly provide a reliable measurement of genome size, and thus a quasi-exhaustive view of the genome architecture. On the other hand, because a gap with C-value is still present, we integrated this metric to correct assembly size estimations to their 'expected C-values'. Similarly, the use of dnaPipeTE allowed us to quantify the repeat content without relying on assembly completeness. In summary, we extensively controlled for the effect of data quality on results and employed methods to minimize it.

## Transposable elements are major contributors to genome size variation

Genome size and TE content have already been reported as tightly linked in eukaryotes (*Elliott and Gregory, 2015*; *Kidwell, 2002*), arthropods (*Sproul et al., 2023*; *Wu and Lu, 2019*), and vertebrates (*Chalopin et al., 2015*). Our results are consistent with this perspective across all the animal species analysed, as well as at the level of ray-finned fishes (*Reinar et al., 2023*), insects (*Heckenhauer et al., 2022*; *Mérel et al., 2025*; *Petersen et al., 2019*; *Sessegolo et al., 2016*), mammals (*Osmanski et al., 2023*), and molluscs (*Martelossi et al., 2023*), strongly indicating TEs as major drivers of genome size variation in metazoans. It should be noted, however, that we mainly focused on some vertebrate groups and insects, while leaving out many animal taxa with fewer genomic resources currently available including much of the animal tree of life, such as most molluscs, annelids, sponges, cnidarians and nematodes. Even for better studied vertebrates, our datasets are far from comprehensive. For instance, the genomes of squamate reptiles are relatively stable in size but show a high variability in repeat content (*Castoe et al., 2011*; *Pasquesi et al., 2018*). A similar case is represented by bird genomes where, according to our analysis and consistently with other studies (*Ji et al., 2022*; *Kapusta and Suh, 2017*), repeat content has a lower capacity to explain size compared to other clades. This could be due to satellites, whose contribution to genome size can be highly variable (*Flynn et al., 2020*; *Pasquesi et al., 2018*; *Peona et al., 2023*). While the remarkable conservation of avian genome sizes has prompted interpretations involving further mechanisms (see discussion below), dnaPipeTE is known to generally underestimate satellite content (*Goubert et al., 2015*). This bias is more relevant for those species that exhibit large fractions of satellites compared to TEs in their repeatome. For instance, the portions of simple and low complexity repeats estimated with dnaPipeTE are consistently smaller than those reported in previous analyses based on assembly annotation for some species, such as *Triatoma infestans* (0.46% *vs* 25%; 7 Mbp *vs* 400 Mbp), *Drosophila eugracilis* (1.28% *vs* 10.89%; 2 Mbp *vs* 25 Mbp), *Drosophila albomicans* (0.06% *vs* 18 to 38%; 0.12 Mbp *vs* 39–85 Mbp), and some other *Drosophila* species (*Pita et al., 2017*; *de Lima and Ruiz-Ruano, 2022*; *Table 3—source data 1*). Although the accuracy of Coevol analyses might occasionally be affected by such underestimations, the effect is likely minimal on the general trends. Inability to detect ancient TE copies is another relevant bias of dnaPipeTE. However, the strong correlation between repeat content and genome size and the consistency of dnaPipeTE and earlGrey results, even in large genomes such as that of *Aedes albopictus*, indicate that dnaPipeTE method is pertinent for our large-scale analysis. Furthermore, such an approach is especially fitting for the examination of recent TEs, as this specific analysis is not biased by very repetitive new TE families that are problematic to assemble.

Another way for genomes to grow involves genomic duplications. Although a high proportion of duplicated BUSCO genes may indicate a low haplotype resolution of the assembly, many species with a high duplication score in our dataset correspond to documented duplication cases, suggesting that such BUSCO statistics may provide an insight into this biological process. However, the contribution of duplicated genes to genome size is minimal compared to the one of TEs, and removing species with high duplication scores does not affect our results: this implies that duplication is unlikely to be the factor causing the relationship between genome size and dN/dS to deviate from the pattern expected from the MHH. Across the animal species considered here, the activity of TEs is therefore a preponderant mechanism of DNA gain, and their evolutionary dynamics appear of prime importance in driving genome size variation.

## Reduced selection efficacy is not associated with increased genome size and TE content

Our dN/dS calculation included several filtering steps by branch length and topology: indeed, selecting markers by such criteria appears to be an essential step to reconcile estimations with different methodologies (*Bastian, 2024*). In addition, our analyses resulted to be robust to species pruning by deviant branch lengths. *Müller et al., 2022* showed that recent $Ne_e$ fluctuations might perturb the expected correlation between long- and short-term estimates of $Ne_e$. According to the nearly neutral theory, alleles that start at low relative frequencies have a mean fixation of ~4Ne generations, under the implicit assumption of constant $N_e$ (*Kimura and Ohta, 1969*). This implies that dN/dS, that accounts for the accumulation of substitutions over time, has a weaker sensitivity to short-term changes in $Ne_e$ compared to estimates based on polymorphism (*Müller et al., 2022*). Additionally, inferences on simulated and empirical data showed that $Ne_e$ changes along branches could be captured and generally recapitulated by dN/dS and LHTs in a framework similar to that of Coevol (*Latrille et al., 2021*). Accordingly, dN/dS assessments by Bio ++ and Coevol are highly concordant between each other and with LHTs. Taken together, our results point at the dN/dS found with the two methods as reliable proxies of long-term $Ne_e$.

If TEs are ascribable to nearly-neutral mutations, a negative correlation of $Ne_e$ with TE expansion, and consequently with genome size – equivalent to a positive association with dN/dS – is expected. However, no such correlations are observed across the sampled species. It is important to note how not treating species traits as non-independent leads to artifactual results (*Figure 4B–C*). For instance, mammals have on average small population sizes and the largest genomes. Conversely, insects tend to have large $Ne_e$ and overall small genomes. With a high sampling power and phylogenetic inertia being taken into account, our meta-analysis clearly points at a phylogenetic structure in the data: the main clades are each confined to separate genome size ranges regardless of their dN/dS variation. The other way around, variability in genome size can be observed in insects, irrespective of their dN/dS. Relying on non phylogenetically corrected models based on a limited number of species (such as that available at the time of the MHH proposal) can thus result in a spurious positive scaling between genome size and $Ne_e$ proxies. To account for a shallower phylogenetic scale, we isolated recently active elements and at the same time explored the same relationships within each clade. Indeed, while the selective effect of elements might be slightly negative as long as they are active, TEs accumulated over long periods of time might be subject to changing dynamics: in the latter case, the pace of sequence erosion could be in the long run independent of drift and lead to different trends of TE retention and degradation in different lineages. Extracting recent elements should thus allow us to have a glimpse of the latest TE colonizations. A positive scaling between the quantity of young TEs and dN/dS found in some cases indicates that relatively recent expansions of TEs could be subject to a more effective negative selection. However, this trend is always very weak and often summarizes that of full TE content within clades. A potential limit of this analysis lies in the application of the same similarity threshold to all species to delimit recent elements. While this is not problematic when comparing species that recently split apart (e.g. *Yang et al., 2024*), some noise might be introduced at large scale, as the quantity of young repeats that evolved on the same time scale can vary according to the mutation rate and generation time of a species.

Interestingly, the correlation patterns between population size proxies and genomic traits emerging within single clades are distinct and sometimes opposite to the expectations of MHH. Mammals display a negative correlation of dN/dS with TE content, a pattern that is uniformly confirmed by LHTs. Not only does this result corroborates previous findings of no relationship between $Ne_e$ and genome size in mammals (*Roddy et al., 2021*), but it supports a correlation opposite to the predictions of the MHH. On the other hand, the observed positive scaling between dN/dS and TE content in ray-finned fishes might lend support to a role of drift on genome size in this clade: this result is also consistent with a previous study which found a negative scaling between genome size and heterozygosity in this group (*Yi and Streelman, 2005*). In birds, population size seems to negatively affect genome size and positively the TE content, a decoupling that is however not surprising given the higher variation of TE load compared to the restricted genome size range. Contrasting signals from the two genomic traits have already been observed by *Ji et al., 2022* who also reported a positive correlation between assembly size and mass, but a negative correlation between TE content and generation time. As previous studies find relatively weak correlations between TE content and genome size in

birds (*Ji et al., 2022*; *Kapusta and Suh, 2017*), it is possible for the very narrow variation of the avian genome sizes to impair the detection of consistent signals. On the other hand, it is conceivable the avian TE diversity to be underappreciated due to the limits of sequencing technologies used so far in resolving complex repeat-rich regions. For instance, employment of long-reads technologies allowed to reveal more extended repeated regions that were previously ignored with short read assemblies (*Kapusta and Suh, 2017*; *Benham et al., 2024*). Besides, quite large fractions might indeed be satellite sequences constituting relevant fractions of the genome that are challenging to identify with reference- or read-based methods (*Edwards et al., 2025*). An 'accordion' dynamic has been proposed whereby higher TE loads are paralleled by equally strong deletional pressures, which could contribute to the maintenance of remarkably small and constant genome sizes in birds, in spite of ongoing TE activity (*Kapusta et al., 2017*; *Kapusta and Suh, 2017*). Finally, the diffused evidence for a positive and a negative correlation of genome size with body mass and metabolic rate, respectively, is also compatible with the adaptationist perspective of powered flight indirectly maintaining small genome sizes in birds as a consequence of the metabolic needs (*Wright et al., 2014*; *Zhang and Edwards, 2012*). In insects, dN/dS scales negatively with genome size, but never with TE content. As eusociality appears to bring about selection relaxation (*Imrit et al., 2020*; *Kapheim et al., 2015*; *Weyna and Romiguier, 2021*), several studies explored the link between $Ne_e$ and genome size in this taxon by focussing on social complexity as a proxy, but with contrasting outcomes: *Mikhailova et al., 2024* find bigger genome size associated with eusociality in Hymenoptera, but the opposite trend in Blattodea; in contrast and partially in accordance with our findings, *Kapheim et al., 2015* and *Koshikawa et al., 2008* report less abundant TEs in eusocial hymenopters and smaller genomes in eusocial termites, respectively. While the approximation of $Ne_e$ based on dN/dS should allow for a quantification of selection efficacy in wider terms than sociality traits, the investigated evolutionary scale might hold an important role in the outcome of such analyses. First, and specifically relative to insects, genome size seems to be subject to different evolutionary pressures – either selective or neutral – within different insect orders (*Cong et al., 2022*), implying that increased drift might not necessarily produce the same effect on genome size across all insect groups. More generally, the five defined clades cover quite different time scales: insects and molluscs have much more ancient origins than mammals and birds, and such distant groups also evolve at very different evolutionary rates, making it difficult to characterize the evolution of their traits on the same evolutionary scale. Nevertheless, the results are still valuable in highlighting the absence of relationship between genetic drift and genome size variation in the long-term evolution of such broad groups, in contrast to previous work focusing at the population level or on recently diverged species (*Cui et al., 2019*; *Mérel et al., 2021*; *Yang et al., 2024*). At the same time, as noted by *Mérel et al., 2025*, comparing very distantly related species – as the insect and molluscan species of our dataset – might overshadow any relationship between genome size and $Ne_e$, either due to dN/dS predicting power being weakened by branch saturation, deep $Ne_e$ fluctuations not being detected by our methods, or to additional factors affecting long-term genome size evolution.

## Do lineage-specific TE dynamics affect genome size evolution?

Our findings do not support the quantity of non-coding DNA being driven in a nearly-neutral fashion by genetic drift. Notably, these results not only reject the theory of extra non-coding DNA being costly for its point mutational risk, but also challenges the more general idea of its accumulation depending on other kinds of detrimental effects, such as increased replication, pervasive transcription, or ectopic recombination. Therefore, our results can be considered more general than a mere rejection of the MHH hypothesis, as they do not support any theory predicting that species with low $Ne_e$ would accumulate more non-coding DNA. In agreement with previous analyses (*Pasquesi et al., 2018*), we find that the proliferation of TEs in particular can, under comparable drift levels, give place to lineage-specific outcomes that mostly do not seem to depend on effective population size. These results contrast with those of other large-scale analyses which instead support the predictions of the drift-barrier hypothesis for a general impact of $Ne_e$ on other genomic features, notably mutation rate (*Bergeron et al., 2023*; *Lynch et al., 2023*; *Wang and Obbard, 2023*) and splicing accuracy (*Bénitière et al., 2024*). To put this in perspective, it should be emphasized that, in the framework of the MHH, the success of nearly-neutral alleles depends on the combination of both $Ne_e$ and liability to mutation of non-coding DNA (*Lynch et al., 2011*). Overall, we studied $Ne_e$ variation without

accounting for the different mutagenic burden posed by non-coding DNA across different lineages. In the case of TEs, inherently assuming the same distribution of selective effect and a constant activity in all species and among TE insertions was assumed. However, it is known that TEs are subject to waves of activity rather than a uniform pace of transposition (*Arkhipova, 2018*). Moreover, given the broad phylogenetic scale of our dataset, it is likely for different levels of hazard to be acting across genomes due to different "host-parasite" dynamics in different animal groups (*Ågren and Wright, 2011*). Such coevolutionary dynamics are, for example, determined by TE silencing mechanisms, which evolve differently across lineages and might influence the degree of genome expansion (*Lechner et al., 2013*; *Zhou et al., 2020*; *Wang et al., 2023*). In general, because of their complex interactions with genomes, TEs are especially likely to deviate from the assumption of gradually mutating sequences. Therefore, treating them as universally slightly deleterious alleles might be an oversimplified model. For instance, while the big genomes of salamanders are not related to small Ne$_e$, the low synonymous substitution rates and low degree of deletions due to ectopic recombination suggest weak mutational hazard of TEs that possibly contributes to the maintenance of genomic gigantism in this group (*Frahry et al., 2015*; *Mohlhenrich and Mueller, 2016*; *Rios-Carlos et al., 2024*). Additionally, lineage-specific TE dynamics themselves might underlie different genomic architectures: for example, mammalian genomes are generally characterized by one preponderant type of active element and by a long-term retention of old TEs (*Osmanski et al., 2023*; *Sotero-Caio et al., 2017*), as in human where a very small proportion of active elements (<0.05%) is unlikely to impose a mutation rate causing genome size variation (*Mills et al., 2007*). Conversely, squamate and teleost fish genomes are smaller and characterized by several, simultaneously active and less abundant TE types (*Duvernell et al., 2004*; *Furano et al., 2004*; *Novick et al., 2009*; *Pasquesi et al., 2018*; *Volff et al., 2003*). These different patterns of genomic organization seem overall associated with different rates of elements' turnover (*Bennett and Riley, 1997*; *Lavoie et al., 2013*; *Novick et al., 2009*; *Volff et al., 2003*). All such variables might alter the selective effect and differentiate TEs from gradually and constantly evolving alleles, eventually contributing to the lack of association between long-term Ne$_e$ and genome size. Finally, *Kapusta et al., 2017* showed that large-scale deletions can be as important as DNA gain in determining genome size, thus questioning the assumption of the rate of elements insertion being greater than their removal rate (*Lynch, 2007*). This implies that the contribution of TEs constitutes just one side of the coin and that deletion bias could also drive the divergence of genome size across lineages, as suggested by several studies linking negatively deletion rates with genome size (*Frahry et al., 2015*; *Ji et al., 2022*; *Kapusta et al., 2017*; *Wang et al., 2014*).

## Perspectives

Evidence for signatures of negative selection against TE proliferation exist at various degrees. In *Anolis* lizards, the ability of TEs to reach fixation can vary between populations of the same species according to population size (*Ruggiero et al., 2017*; *Tollis and Boissinot, 2013*). Furthermore, Ne$_e$ was found to negatively correlate with genome size and TE expansion at the intraspecific level in *Drosophila suzukii* (*Mérel et al., 2021*) and at the interspecific level in fruit flies (*Mérel et al., 2025*), asellid isopods (*Lefébure et al., 2017*), and killifishes (*Cui et al., 2019*), supporting the role of genetic drift in determining recent differences in genome size among closely related animal species. Given the very different taxonomic scale of such works and ours, and with the perspective of lineage-specific interaction between genome and genomic parasites in mind, our negative results for the MHH at metazoan scale are not incompatible with an effect of Ne$_e$ on genome size within specific clades. In a nutshell, although an increase in genetic drift seems to lead to the short-term accumulation of transposable elements, this process is not visible in the long term, suggesting that it fades over time. A general mechanism of selection preventing the proliferation of non-coding DNA and TEs in animals might exist but its results be detectable only at a sufficiently short evolutionary time. In this sense, the lack of evidence for MHH in other clade-specific studies might be due to the methodological challenges of either estimating a suitable marker of Ne$_e$ or investigating too distantly related lineages. Moreover, the contrasting outcomes of such studies might reflect an actual variability in the selective effect of TEs not compatible with a general selection mechanism. Further reducing the phylogenetic scale under study and systematically exploring the consequences of Ne$_e$ variation within independent biological systems could therefore provide an alternative way to test the impact of drift, while removing the confounding effects due to different genomic backgrounds.

## Methods

### Dataset

All the metazoan reference assemblies available as of November 14th 2021 were used, except for insect genomes which were drawn from *Sproul et al., 2023*, for a total of 3,214 assemblies. For each assembly, quality metrics were computed with Quast 5.0.2 (*Gurevich et al., 2013*) and genome completeness was assessed with BUSCO 5.2.2 using the 954 markers of the metazoa_odb10 geneset (*Supplementary file 1*). Availability of raw reads was verified with SRA Explorer (https://github.com/ewels/sra-explorer; *Ewels et al., 2025*). All the assemblies with either a contig N50 smaller than 50 kb, less than 70% of complete BUSCO orthologs, or without available reads were excluded from TE and dN/dS analyses. The subdivision into Actinopteri (N=148), Aves (N=260), Insecta (N=189), Mammalia (N=182), Mollusca (N=28) was adopted to perform alignments, phylogenies, dN/dS estimation with Bio ++ and Coevol runs (see below).

### Genome size estimation

Assembly sizes and C-values were jointly used to estimate genome size. C-values measured by either flow cytometry (FCM), Feulgen densitometry (FD) or Feulgen image analysis densitometry (FIA) were collected from https://www.genomesize.com/ (last accessed 6 october 2022) for all available species of our initial dataset with contig N50 ≥ 50 kb, totalling 465 measurements for 365 species (*Figure 2 - source data 1*). To assign a unique C-value, when multiple values were present for one species, the most recent one was retained and, if dates were the same, the average was used. For all the other species having contig N50 ≥ 50 kb but with no available C-value record, genome size was calculated as an expected C value predicted from a WLS where the 465 FCM, FD and FIA estimations were the independent variables (for details see https://github.com/albmarino/Meta-analysis_scripts, copy archived at *Marino, 2024*). Out of all the records in this training dataset for genome size, 93 correspond to ray-finned fishes, 93 to mammals, 92 to birds, 106 to insects, and 9 to molluscs, overall mirroring the taxa represented in the final dataset. For the purpose of our analysis, C-values were used for the species for which such data were available, while the expected C-value was used as genome size estimation in all the other cases, regardless of the type of sequencing data used for the assembly (*Supplementary file 1*; *Figure 2—source data 1*).

### Gene alignment

The 954 annotated single-copy BUSCO genes were aligned with the pipeline OMM_MACSE 11.05 using MACSE 2.06 (*Ranwez et al., 2018*; *Scornavacca et al., 2019*). Alignments were performed separately within each clade - Actinopteri, Aves, Insecta, Mammalia, and Mollusca.

### Phylogeny

Phylogenies were computed separately for each clade with IQ-TREE 1.6.12 (*Nguyen et al., 2015*). JTT +F + R10 substitution model was selected with ModelFinder (-m MFP option; *Kalyaanamoorthy et al., 2017*). For reasons of computing power and time, we have reconstructed the phylogenies of each clade independently and then grouped them together to create a single complete phylogenetic tree (see below). The same set of 107 concatenated BUSCO amino-acid sequences was used to calculate all the phylogenies. However, since this produced a spurious relationship in the mammalian tree with paraphyly of primates, an alternative set of randomly selected 100 genes was used instead for the phylogeny of Mammalia (*Supplementary file 2*). Each phylogeny was rooted using an outgroup species belonging to its respective sister clade: the outgroup sequences were added to gene alignments with the enrichAlignment function from MACSE; the outgroup +clade gene alignments were concatenated and used to recompute the outgroup +clade phylogeny taking into account the previously computed tree topology of the clade. The outgroups were then removed, and rooted clade phylogenies were merged together manually using the tree editor program Baobab (*Dutheil and Galtier, 2002*). Finally, 50 top-shared genes across all species were chosen among the set of 107 genes (*Supplementary file 2*) to recalculate the branch lengths of the whole dataset phylogeny: with the MACSE program alignTwoProfiles the nucleotide gene alignment of one clade was joined to the one of its respective sister clade until achievement of the whole dataset alignment. Branch lengths were then estimated based on the 50-genes concatenate and the tree topology (for the detailed workflow, see https://github.com/albmarino/Meta-analysis_scripts; *Marino, 2024*).

## dN/dS estimation

When genetic drift is strong, slightly deleterious mutations are more likely to reach fixation than under conditions of high $Ne_e$ and more effective selection (*Ohta, 1992*). The genome-wide fixation rate of non-synonymous mutations is expected to drive the dN/dS ratio due to nearly-neutral mutations responding to different $Ne_e$: a higher dN/dS accounts for more frequent accumulation of mildly deleterious mutations over time due to small $Ne_e$, while lower dN/dS is associated with a stronger effect of selection against slightly deleterious non-synonymous mutations due to high $Ne_e$ (*James et al., 2016*; *Romiguier et al., 2014*; *Weyna and Romiguier, 2021*; *Woolfit and Bromham, 2005*). This is also supported at the polymorphism level, with higher pN/pS and accumulation of slightly deleterious mutations in smaller populations (*Leroy et al., 2021*; *Dussex et al., 2023*).

Before dN/dS calculation, sequences with more than 10% of their length occupied by insertions were preemptively removed from BUSCO alignments. Estimation of dN/dS on either very long or short terminal branches might lead to loss of accuracy due to branch saturation (*Weber et al., 2014*) or to a higher variance of substitution rates, respectively. Furthermore, shared polymorphism can be captured in the substitution rates when closely related species are compared, and further contribute to bias dN/dS (*Mugal et al., 2020*). To correct for such issues, genes with deviant topology were identified and removed from every clade with PhylteR using default parameters (*Comte et al., 2023*). Moreover, genes exhibiting branch lengths shorter than 0.001, for which dN/dS could have a large variance, were also not integrated in the dN/dS calculation of a species.

We then used bppml and mapnh from the Bio ++ libraries to estimate dN/dS on terminal branches (*Dutheil et al., 2006*; *Guéguen et al., 2013*; *Guéguen and Duret, 2018*; *Romiguier et al., 2012*). bppml calculates the parameters under a homogenous codon model YN98 (F3X4). Next, mapnh maps substitutions along the tree branches and estimates dN and dS. More precisely, the substitution rate is given by the number of substitutions mapped according to the model parameters normalized by the number of substitutions of the same category (i.e. synonymous, non-synonymous) that would occur under the same neutral model (*Bolívar et al., 2019*). Therefore, dN and dS are calculated for each species as follows:

$$\frac{\sum_{i=1}^{n} K\left(i\right)}{\sum_{i=1}^{n} \dfrac{O\left(i\right)}{l\left(i\right)}}$$

where *n* is the number of genes, *K* is the substitutions count as mapped by the substitution model calculated for the gene, *O* is the substitutions count as mapped under the same neutral model, and *l* is the branch length of the given species for that gene. In addition to the gene filtering described above, Bio ++ dN/dS was recalculated on a reduced dataset where the longest (>1) and shortest (<0.01) branches were removed, in order to ensure that substitution saturation and segregating polymorphism did not influence the results. Terminal branches with more than 1 and less than 0.01 amino-acid substitutions per site were removed, and dN/dS was recalculated on the trimmed phylogenies with the same method described above.

The same metric was estimated with Coevol 1.6 (*Lartillot and Poujol, 2011*). Coevol models the co-evolution of dN/dS and continuous traits along branches following a multivariate Brownian diffusion process, thus reducing the variance in the dN/dS of the smallest branches (*Brevet and Lartillot, 2021*; *Lartillot and Delsuc, 2012*). Bio ++ dN/dS was therefore compared with dN/dS estimated by Coevol on terminal branches to verify the consistency between the two methods.

## Compilation of life history traits

LHTs – body mass, longevity, generation time, among others – are found to be related to $Ne_e$ in mammals, birds, and amniotes in general (*Bolívar et al., 2019*; *Figuet et al., 2016*; *Nikolaev et al., 2007*; *Popadin et al., 2007*). Available LHTs were assigned to the species of our dataset using information from several resources. Adult body mass, body length, maximum longevity, basal metabolic rate, age at first birth, population siz,e and population density were assigned to mammalian species using PanTHERIA (*Jones et al., 2009*). For birds, body mass information was extracted from *Dunning, 2007*. Shallow to deep depth range, longevity in the wild, body lengt,h and body mass were compiled for ray-finned fishes from https://www.fishbase.org/ using the rfishbase R package (*Boettiger et al.,*

*2012*). Additionally, age at sexual maturity, adult body mass, maximum longevit,y and metabolic rate were extracted from AnAge (*Tacutu et al., 2018*), as well as body mass and metabolic rate from AnimalTraits (*Herberstein et al., 2022*): such data were used to complement information when missing from the databases cited above. All the retrieved LHTs and their relative source are reported in *Table 2—source data 1*.

## TE quantification

TEs were annotated with a pipeline employing dnaPipeTE (*Goubert et al., 2015*) in two rounds (https://github.com/sigau/pipeline_dnapipe, copy archived at *Debaecker and Marino, 2025*). Raw reads are filtered with UrQt (*Modolo and Lerat, 2015*) or fastp (*Chen et al., 2018*) and undergo a first 'dirty' dnaPipeTE round. The obtained dnaPipeTE contigs are mapped against a database of organellar, fungal, bacterial, and archaean reference sequences with Minimap2 (*Li, 2018*), and the original reads matching contaminant sequences are removed with SAMtools (*Danecek et al., 2021*). Finally, the quality- and contaminant-filtered reads are used to perform a second 'clean' dnaPipeTE round. dnaPipeTE was configured with the Dfam 3.5 and RepBaseRepeatMaskerEdition-20181026 repeat libraries and was run with a genome coverage of 0.25.

To verify the consistency of dnaPipeTE estimations, the dnaPipeTE-based pipeline was benchmarked on a subset of 29 dipteran species against EarlGrey 1.3, an automated pipeline performing TE annotation on genome assemblies (*Baril et al., 2024*). EarlGrey was configured with the same libraries as dnaPipeTE and was run with 'metazoa' as the search term.

The results of the second dnaPipeTE round were used to extrapolate the total and recent TE content, the latter being defined by all the reads below 5% of divergence from the corresponding consensus. The two contents were extracted by adapting dnaPT_landscapes.sh from the dnaPT_utils repository (https://github.com/clemgoub/dnaPT_utils; *Goubert, 2022*) to a custom R script (https://github.com/albmarino/Meta-analysis_scripts; *Marino, 2024*).

## Gene duplication

To account for the effect of whole genome or segmental duplications, we used the BUSCO Duplicated score: if big-scale duplication events recently took place, a higher score should be observed genomewide even for conserved genes. As many of the genomes with BUSCO Duplicated score above 30% corresponded to reported cases of genomic duplication, we used this threshold to perform PIC analysis with and without species whose genome size is potentially more affected by duplication events.

## Phylogenetic independent contrasts and Coevol reconstruction

The correlations of Bio ++ and Coevol dN/dS with LHTs, as well as with genome size and TE content, were tested with PIC to correct for the covariation of traits due to the phylogenetic relatedness of species (*Felsenstein, 1985*). PICs were performed on the whole dataset, the trimmed dataset, and within every clade with the R packages ape 5.7.1 (*Paradis et al., 2004*), nlme 3.1.162, and caper 1.0.1. Results were plotted with the ggplot2 3.4.2 package. Additionally, Coevol 1.6 was run to test for the coevolution of dN/dS and traits: sequence substitution processes and quantitative traits such as LHTs, genome size, and TE content are here assumed to covary along the phylogeny as a multivariate Brownian motion process. Coevol infers trait values on internal nodes and terminal branches (those used for PIC), as well as correlation coefficients and their relative posterior probabilities. Due to computational limitations, Coevol analysis was carried out separately on every clade and on a limited number of genes. Genes were selected according to their GC content at the third codon position (GC3). Indeed, mixing genes with heterogeneous base composition in the same concatenate might result in an alteration of the calculation of codon frequencies, and consequently impair the accuracy of the model estimating substitution rates (*Mérel et al., 2025*). Moreover, genes with different GC3 levels can reflect different selective pressures, as highly expressed genes should be enriched in optimal codons as a consequence of selection on codon usage. In *Drosophila*, where codon usage bias is at play, most optimal codons present G/C bases at the third position (*Duret and Mouchiroud, 1999*), meaning that genes with high GC3 content should evolve under stronger purifying selection than GC3-poor genes. Accordingly, *Mérel et al., 2025* do find a stronger relationship between dN/dS and genome size when using GC3-poor genes, as compared to GC3-rich genes or gene concatenates of random GC3 composition. Finally, dN/dS can be influenced by GC-biased

gene conversion (*Bolívar et al., 2019*; *Ratnakumar et al., 2010*), and the strength at which such substitution bias acts can be reflected by base composition. For these reasons, two sets of 50 genes with similar GC3 content were defined in order to employ genes undergoing similar evolutionary regimes. Markers and species were subject to more stringent filtering in order to have as much information as possible for each species. 16 species with less than 50% of single-copy orthologs were further filtered out from the dataset. In addition to the PhylteR step, only genes represented in at least 95% of the species of a clade were retained. From those, the 50 GC3-poorest and the 50 GC3-richest genes were chosen. Coevol was then run with both the gene sets for every clade. Convergence of the MCMC chains was checked visually by plotting the evolution of statistics. Likelihood values and correlations were estimated, running the chains for a minimum of 1000 steps and discarding the first 400 steps as burn-in.

## Acknowledgements

We are grateful to Laurent Duret, Nicolas Lartillot, Tristan Lefébure, Mélodie Bastian and Florian Bénitière for the helpful discussions. We thank Mélodie Bastian and Florian Bénitière for methodological exchanges to ensure to have a dN/dS accurately estimated. We thank Laurent Guéguen for his help with dN/dS analyses. This work was performed using the computing facilities of the Montpellier Bioinformatics Biodiversity platform (MBB) and CC LBBE/PRABI. We are grateful to three anonymous reviewers whose comments helped improve the quality of this manuscript. A CC-BY public copyright licence has been applied by the authors to the present document and will be applied to all subsequent versions up to the Author Accepted Manuscript arising from this submission, in accordance with the grant's open access conditions.

## Additional information

### Funding

| Funder | Grant reference number | Author |
|---|---|---|
| Agence Nationale de la Recherche | ANR-20-CE02-0008-01 "NeGA" | Alba Marino Gautier Debaecker Anna-Sophie Fiston-Lavier Annabelle Haudry Benoit Nabholz |

The funders had no role in study design, data collection and interpretation, or the decision to submit the work for publication.

### Author contributions

Alba Marino, Conceptualization, Formal analysis, Investigation, Writing – original draft, Writing – review and editing; Gautier Debaecker, Software, Formal analysis, Writing – review and editing; Anna-Sophie Fiston-Lavier, Annabelle Haudry, Conceptualization, Supervision, Funding acquisition, Validation, Writing – review and editing; Benoit Nabholz, Conceptualization, Supervision, Funding acquisition, Validation, Project administration, Writing – review and editing

### Author ORCIDs

Alba Marino ⓘ https://orcid.org/0009-0005-0984-2524
Anna-Sophie Fiston-Lavier ⓘ https://orcid.org/0000-0002-7306-6532
Annabelle Haudry ⓘ https://orcid.org/0000-0001-6088-0909
Benoit Nabholz ⓘ https://orcid.org/0000-0003-0447-1451

Reviewer #1 (Public review): https://doi.org/10.7554/eLife.100574.3.sa1
Reviewer #3 (Public review): https://doi.org/10.7554/eLife.100574.3.sa2
Author response https://doi.org/10.7554/eLife.100574.3.sa3

## Additional files

### Supplementary files

Supplementary file 1. Metadata, assembly metrics, BUSCO scores and genome sizes for the initial 3214 species. C-values and expected C-values are reported only for species with Quast_ContigN50≥50 kb. Method and Notes_cvalue_method report the method used for genome size measurement (FCM = Flow Cytometry, FD = Feulgen Densitometry, FIA = Feulgen Image Analysis) and how C-value was chosen from https://www.genomesize.com/, respectively. Expected C-values are the C-values predicted from the WLS trained on the dataset in *Figure 2—source data 1*. C-values are employed as genome size for the species with a record in the Genome Size database, while the expected C-values are used for the species without a record.

Supplementary file 2. BUSCO genes used to calculate the clade phylogenies and the branch lengths of the whole tree.

MDAR checklist

### Data availability

The current manuscript is a computational study, so no data have been generated. All used data can be freely accessed from the identifiers and sources provided in *Supplementary file 1* and *Table 2—source data 1*. Detailed commands and custom scripts are available at GitHub, copy archived at *Marino, 2024*.

The following previously published dataset was used:

| Author(s) | Year | Dataset title | Dataset URL | Database and Identifier |
|---|---|---|---|---|
| Jones KE, Bielby J, Cardillo M | 2016 | PanTHERIA | https://doi.org/10.6084/m9.figshare.3531875.v1 | figshare, 10.6084/m9.figshare.3531875.v1 |

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
