## [Editor Report · eLife Assessment]

This **important** study offers a powerful empirical test of a highly influential hypothesis in population genetics. It incorporates a large number of animal genomes spanning a broad phylogenetic spectrum and treats them in a rigorous unified pipeline, providing the **convincing** negative result that effective population size scales neither with the content of transposable elements nor with overall genome size. These observations demonstrate that there is still no simple, global hypothesis that can explain the observed variation in transposable element content and genome size in animals.

---

## [Referee Report · Reviewer #1 (Public review)]

Summary:

One enduring mystery involving the evolution of genomes is the remarkable variation they exhibit with respect to size. Much of that variation is due to differences in the number of transposable elements, which often (but not always) correlates with the overall quantity of DNA. Amplification of TEs is nearly always either selectively neutral or negative with respect to host fitness. Given that larger effective population sizes are more efficient at removing these mutations, it has been hypothesized that TE content, and thus overall genome size, may be a function of effective population size. The authors of this manuscript test this hypothesis by using a uniform approach to analysis of several hundred animal genomes, using the ration of synonymous to nonsynonymous mutations in coding sequence as a measure of overall strength of purifying selection, which serves as a proxy for effective population size over time. The data convincingly demonstrates that it is unlikely that effective population size has a strong effect on TE content and, by extension, overall genome size (except for birds, which are weird).

Strengths:

Although this ground has been covered before in many other papers, the strength of this analysis is that it is comprehensive and treats all the genomes with the same pipeline, making comparisons more convincing. Although this is a negative result, it is important because it is relatively comprehensive and indicates that there will be no simple, global hypothesis that can explain the observed variation.

Weaknesses:

In the first draft, the authors slipped between assertions of correlation and assertions of cause-effect relationships not established in the results. However, they have corrected the language so that it more carefully makes this distinction.

---

## [Referee Report · Reviewer #3 (Public review)]

The Mutational Hazard Hypothesis (MHH) suggests that lineages with smaller effective population sizes should accumulate slightly deleterious transposable elements leading to larger genome size. Marino and colleagues tested the MHH using a set of 807 vertebrate, mollusc and insect species. The authors mined repeats de novo and estimated dN/dS for each genome. Then, they used dN/dS and life history traits as reliable proxies for effective population size and tested for correlations between these proxies and repeat content while accounting for phylogenetic nonindependence. The results suggest that overall, lineages with lower effective population sizes do not exhibit increases in repeat content or genome size. This contrasts with expectations from the MHH. The authors speculate that changes in genome size may be driven by lineage-specific host-TE conflicts rather than effective population size.

Strengths:

The general conclusions of this paper are supported by a powerful dataset of phylogenetically diverse species. Furthermore, the hypothesis tested is important and has proved challenging to test in the past due to technical challenges and confounding factors. The use of C-values rather than assembly size for many species (when available) helps to mitigate the challenges associated with underrepresentation of repetitive regions in short-read based genome assemblies. Overall, both the phylogenetic breadth of species considered and the approaches employed make the results highly convincing.

Weaknesses:

My primary concerns were related to possible biases in the author's data due to their approach to TE annotation. The authors have sufficiently acknowledged and addressed these concerns in their revised manuscript. I note no further weaknesses.

---

## [Author Response]

The following is the authors’ response to the original reviews.

**Public Reviews:**

**Reviewer #1 (Public review):**
Summary:One enduring mystery involving the evolution of genomes is the remarkable variation they exhibit with respect to size. Much of that variation is due to differences in the number of transposable elements, which often (but not always) correlates with the overall quantity of DNA. Amplification of TEs is nearly always either selectively neutral or negative with respect to host fitness. Given that larger effective population sizes are more efficient at removing these mutations, it has been hypothesized that TE content, and thus overall genome size, may be a function of effective population size. The authors of this manuscript test this hypothesis by using a uniform approach to analysis of several hundred animal genomes, using the ratio of synonymous to nonsynonymous mutations in coding sequence as a measure of the overall strength of purifying selection, which serves as a proxy for effective population size over time. The data convincingly demonstrates that it is unlikely that effective population size has a strong effect on TE content and, by extension, overall genome size (except for birds).Strengths:Although this ground has been covered before in many other papers, the strength of this analysis is that it is comprehensive and treats all the genomes with the same pipeline, making comparisons more convincing. Although this is a negative result, it is important because it is relatively comprehensive and indicates that there will be no simple, global hypothesis that can explain the observed variation.Weaknesses:In several places, I think the authors slip between assertions of correlation and assertions of cause-effect relationships not established in the results.

Several times in the previous version of the manuscript we used the expression “effect of dN/dS on…” which might suggest a causal relationship. We have rephrased these expressions and highlighted the changes in the main text, so that correlation is not mistaken with causation (see also responses to detailed comments below).

In other places, the arguments end up feeling circular, based, I think, on those inferred causal relationships. It was also puzzling why plants (which show vast differences in DNA content) were ignored altogether.

The analysis focuses on metazoans for two reasons: one practical and one fundamental.

The practical reason is computational. Our analysis included TE annotation, phylogenetic estimation and dN/dS estimation, which would have been very difficult with the hundreds, if not thousands, of plant genomes available. If we had included plants, it would have been natural to include fungi as well, to have a complete set of multicellular eukaryotic genomes, adding to the computational burden. The second fundamental reason is that plants show important genome size differences due to more frequent whole genome duplications (polyploidization) than in animals. It is therefore possible that the effect of selection on genome size is different in these two groups, which would have led us to treat them separately, decreasing the interest of this comparison. For these reasons we chose to focus on animals that still provide very wide ranges of genome size and population size well suited to test the impact of genetic drift on the genomic TE content.

**Reviewer #2 (Public review):**
Summary:The Mutational Hazard Hypothesis (MHH) is a very influential hypothesis in explaining the origins of genomic and other complexity that seem to entail the fixation of costly elements. Despite its influence, very few tests of the hypothesis have been offered, and most of these come with important caveats. This lack of empirical tests largely reflects the challenges of estimating crucial parameters.The authors test the central contention of the MHH, namely that genome size follows effective population size (Ne). They martial a lot of genomic and comparative data, test the viability of their surrogates for Ne and genome size, and use correct methods (phylogenetically corrected correlation) to test the hypothesis. Strikingly, they not only find that Ne is not THE major determinant of genome size, as is argued by MHH, but that there is not even a marginally significant effect. This is remarkable, making this an important paper.Strengths:The hypothesis tested is of great importance.The negative finding is of great importance for reevaluating the predictive power of the tested hypothesis.The test is straightforward and clear.The analysis is a technical tour-de-force, convincingly circumventing a number of challenges of mounting a true test of the hypothesis.Weaknesses:I note no particular strengths, but I believe the paper could be further strengthened in three major ways.(1) The authors should note that the hypothesis that they are testing is larger than the MHH.The MHH hypothesis says that(i) low-Ne species have more junk in their genomes and(ii) this is because junk tends to be costly because of increased mutation rate to nulls, relative to competing non/less-junky alleles.The current results reject not just the compound (i+ii) MHH hypothesis, but in fact any hypothesis that relies on i. This is notably a (much) more important rejection. Indeed, whereas MHH relies on particular constructions of increased mutation rates of varying plausibility, the more general hypothesis i includes any imaginable or proposed cost to the extra sequence (replication costs, background transcription, costs of transposition, ectopic expression of neighboring genes, recombination between homologous elements, misaligning during meiosis, reduced organismal function from nuclear expansion, the list goes on and on). For those who find the MHH dubious on its merits, focusing this paper on the MHH reduces its impact - the larger hypothesis that the small costs of extra sequence dictate the fates of different organisms' genomes is, in my opinion, a much more important and plausible hypothesis, and thus the current rejection is more important than the authors let on.

The MHH is arguably the most structured and influential theoretical framework proposed to date based on the null assumption (i), therefore setting the paper up with the MHH is somehow inevitable. Because of this, we mostly discuss the assumption (ii) (the mutational aspect brought about by junk DNA) and the peculiarities of TE biology that can drive the genome away from the expectations of (i). We however agree that the hazard posed by extra DNA is not limited to the gain of function via the mutation process, but can be linked to many other molecular processes as mentioned above. Moreover, we also agree that our results can be interpreted within the general framework of the nearly-neutral theory. They demonstrate that mutations, whether increasing or decreasing genome size, have a distribution of fitness effects that falls outside the range necessary for selection in larger populations. In the revised manuscript, we made the concept of hazard more comprehensive and further stressed that this applies not only to TEs but any nearly-neutral mutation affecting non-coding DNA (lines 491-496): “Notably, these results not only reject the theory of extra non-coding DNA being costly for its point mutational risk, but also challenges the more general idea of its accumulation depending on other kinds of detrimental effects, such as increased replication, pervasive transcription, or ectopic recombination. Therefore, our results can be considered more general than a mere rejection of the MHH hypothesis, as they do not support any theory predicting that species with low Ne would accumulate more non-coding DNA.”

(2) In addition to the authors' careful logical and mathematical description of their work, they should take more time to show the intuition that arises from their data. In particular, just by looking at Figure 1b one can see what is wrong with the non-phylogenetically-corrected correlations that MHH's supporters use. That figure shows that mammals, many of which have small Ne, have large genomes regardless of their Ne, which suggests that the coincidence of large genomes and frequently small Ne in this lineage is just that, a coincidence, not a causal relationship. Similarly, insects by and large have large Ne, regardless of their genome size. Insects, many of which have large genomes, have large Ne regardless of their genome size, again suggesting that the coincidence of this lineage of generally large Ne and smaller genomes is not causal. Given that these two lineages are abundant on earth in addition to being overrepresented among available genomes (and were even more overrepresented when the foundational MHH papers collected available genomes), it begins to emerge how one can easily end up with a spurious non-phylogenetically corrected correlation: grab a few insects, grab a few mammals, and you get a correlation. Notably, the same holds for lineages not included here but that are highly represented in our databases (and all the more so 20 years ago): yeasts related to *S. cerevisiae* (generally small genomes and large median Ne despite variation) and angiosperms (generally large genomes (compared to most eukaryotes) and small median Ne despite variation). Pointing these clear points out will help non-specialists to understand why the current analysis is not merely a they-said-them-said case, but offers an explanation for why the current authors' conclusions differ from the MHH's supporters and moreover explain what is wrong with the MHH's supporters' arguments.

We thank the referee for this perspective. We agree that comparing dispersion of the points from the non-phylogenetically corrected correlation with the results of the phylogenetic contrasts intuitively emphasizes the importance of accounting for species relatedness. We added on to the discussion to stress the phylogenetic structure present in the data (lines 408-417): “It is important to note how not treating species traits as non-independent leads to artifactual results (Figure 2B-C). For instance, mammals have on average small population sizes and the largest genomes. Conversely, insects tend to have large Ne and overall small genomes. With a high sampling power and phylogenetic inertia being taken into account, our meta-analysis clearly points at a phylogenetic structure in the data: the main clades are each confined to separate genome size ranges regardless of their dN/dS variation. The other way around, variability in genome size can be observed in insects, irrespective of their dN/dS. Relying on non phylogenetically corrected models based on a limited number of species (such as that available at the time of the MHH proposal) can thus result in a spurious positive scaling between genome size and Ne proxies.”

(3) A third way in which the paper is more important than the authors let on is in the striking degree of the failure of MHH here. MHH does not merely claim that Ne is one contributor to genome size among many; it claims that Ne is THE major contributor, which is a much, much stronger claim. That no evidence exists in the current data for even the small claim is a remarkable failure of the actual MHH hypothesis: the possibility is quite remote that Ne is THE major contributor but that one cannot even find a marginally significant correlation in a huge correlation analysis deriving from a lot of challenging bioinformatic work. Thus this is an extremely strong rejection of the MHH. The MHH is extremely influential and yet very challenging to test clearly. Frankly, the authors would be doing the field a disservice if they did not more strongly state the degree of importance of this finding.

We respectfully disagree with the review that there is currently no evidence for an effect of Ne on genome size evolution. While it is accurate that our large dataset allows us to reject the universality of Ne as the major contributor to genome size variation, this does not exclude the possibility of such an effect in certain contexts. Notably, there are several pieces of evidence that find support for Ne to determine genome size variation and to entail nearly-neutral TE dynamics under certain circumstances, e.g. of particularly strongly contrasted Ne and moderate divergence times (Lefébure et al., 2017 Genome Res 27: 1016-1028; Mérel et al., 2021 Mol Biol Evol 38: 4252-4267; Mérel et al., 2024 biorXiv: 2024-01; Tollis and Boissinot, 2013 Genome Biol Evol 5: 1754-1768; Ruggiero et al., 2017 Front Genet 8: 44). The strength of such works is to analyze the short-term dynamics of TEs in response to N_e_ within groups of species/populations, where the cost posed by extra DNA is likely to be similar. Indeed, the MHH predicts genome size to vary according to the combination of drift and mutation under the nearly-neutral theory of molecular evolution. Our work demonstrates that it is not true universally but does not exclude that it could exist locally. Moreover, defence mechanisms against TEs proliferation are often complex molecular machineries that might or might not evolve according to different constraints among clades. We have detailed these points in the discussion (lines 503-518).

**Reviewer #3 (Public review):**
SummaryThe Mutational Hazard Hypothesis (MHH) suggests that lineages with smaller effective population sizes should accumulate slightly deleterious transposable elements leading to larger genome sizes. Marino and colleagues tested the MHH using a set of 807 vertebrate, mollusc, and insect species. The authors mined repeats de novo and estimated dN/dS for each genome. Then, they used dN/dS and life history traits as reliable proxies for effective population size and tested for correlations between these proxies and repeat content while accounting for phylogenetic nonindependence. The results suggest that overall, lineages with lower effective population sizes do not exhibit increases in repeat content or genome size. This contrasts with expectations from the MHH. The authors speculate that changes in genome size may be driven by lineage-specific host-TE conflicts rather than effective population size.StrengthsThe general conclusions of this paper are supported by a powerful dataset of phylogenetically diverse species. The use of C-values rather than assembly size for many species (when available) helps mitigate the challenges associated with the underrepresentation of repetitive regions in short-read-based genome assemblies. As expected, genome size and repeat content are highly correlated across species. Nonetheless, the authors report divergent relationships between genome size and dN/dS and TE content and dN/dS in multiple clades: Insecta, Actinopteri, Aves, and Mammalia. These discrepancies are interesting but could reflect biases associated with the authors' methodology for repeat detection and quantification rather than the true biology.WeaknessesThe authors used dnaPipeTE for repeat quantification. Although dnaPipeTE is a useful tool for estimating TE content when genome assemblies are not available, it exhibits several biases. One of these is that dnaPipeTE seems to consistently underestimate satellite content (compared to repeat masker on assembled genomes; see Goubert et al. 2015). Satellites comprise a significant portion of many animal genomes and are likely significant contributors to differences in genome size. This should have a stronger effect on results in species where satellites comprise a larger proportion of the genome relative to other repeats (e.g. *Drosophila* virilis, >40% of the genome (Flynn et al. 2020); Triatoma infestans, 25% of the genome (Pita et al. 2017) and many others). For example, the authors report that only 0.46% of the Triatoma infestans genome is "other repeats" (which include simple repeats and satellites). This contrasts with previous reports of {greater than or equal to}25% satellite content in Triatoma infestans (Pita et al. 2017). Similarly, this study's results for "other" repeat content appear to be consistently lower for *Drosophila* species relative to previous reports (e.g. de Lima & Ruiz-Ruano 2022). The most extreme case of this is for *Drosophila* albomicans where the authors report 0.06% "other" repeat content when previous reports have suggested that 18%->38% of the genome is composed of satellites (de Lima & Ruiz-Ruano 2022). It is conceivable that occasional drastic underestimates or overestimates for repeat content in some species could have a large effect on coevol results, but a minimal effect on more general trends (e.g. the overall relationship between repeat content and genome size).

There are indeed some discrepancies between our estimates of low complexity repeats and those from the literature due to the approach used. Hence, occasional underestimates or overestimates of repeat content are possible. As noted, the contribution of “Other” repeats to the overall repeat content is generally very low, meaning an underestimation bias. We thank the reviewer for providing this interesting review.

We emphasized these points in the discussion of our revised manuscript (lines 358-376): “While the remarkable conservation of avian genome sizes has prompted interpretations involving further mechanisms (see discussion below), dnaPipeTE is known to generally underestimate satellite content (Goubert et al. 2015). This bias is more relevant for those species that exhibit large fractions of satellites compared to TEs in their repeatome. For instance, the portions of simple and low complexity repeats estimated with dnaPipeTE are consistently smaller than those reported in previous analyses based on assembly annotation for some species, such as *Triatoma infestans* (0.46% vs 25%; 7 Mbp vs 400 Mbp), *Drosophila eugracilis* (1.28% vs 10.89%; 2 Mbp vs 25 Mbp), *Drosophila albomicans* (0.06% vs 18 to 38%; 0.12 Mbp vs 39 to 85 Mbp) and some other *Drosophila* species (Pita et al. 2017; de Lima and Ruiz-Luano 2022; Supplemental Table S2). Although the accuracy of Coevol analyses might occasionally be affected by such underestimations, the effect is likely minimal on the general trends. Inability to detect ancient TE copies is another relevant bias of dnaPipeTE. However, the strong correlation between repeat content and genome size and the consistency of dnaPipeTE and earlGrey results, even in large genomes such as that of *Aedes albopictus*, indicate that dnaPipeTE method is pertinent for our large-scale analysis. Furthermore, such an approach is especially fitting for the examination of recent TEs, as this specific analysis is not biased by very repetitive new TE families that are problematic to assemble.”

Not being able to correctly estimate the quantity of satellites might pose a problem for quantifying the total content of junk DNA. However, the overall repeat content mostly composed of TEs correlates very well with genome size, both in the overall dataset and within clades (with the notable exception of birds) so we are confident that this limitation is not the explanation of our negative results. Moreover, while satellite information might be missing, this is not problematic to test our hypothesis, as we focus on TEs, whose proliferation mechanism differs significantly from that of tandem repeats and largely account for genome size variation.

Another bias of dnaPipeTE is that it does not detect ancient TEs as well as more recently active TEs (Goubert et al., 2015 Genome Biol Evol 7: 1192-1205). Thus, the repeat content used for PIC and coevolve analyses here is inherently biased toward more recently inserted TEs. This bias could significantly impact the inference of long-term evolutionary trends.

Indeed, dnaPipeTE is not good at detecting old TE copies due to the read-based approach, biasing the outcome towards new elements. We agree that TE content can be underestimated, especially in those genomes that tend to accumulate TEs rather than getting rid of them. However, the sum of old TEs and recent TEs is extremely well correlated to genome size (Pearson’s correlation: r = 0.87, p-value < 2.2e-16; PIC: slope = 0.22, adj-R^2^ = 0.42, p-value < 2.2e-16). Our main result therefore does not rely on an accurate estimation of old TEs. In contrast, we hypothesized that recent TEs could be interesting because selection could be more likely to act on TEs insertion and dynamics rather than on non-coding DNA as a whole. Our results demonstrate that this is not the case. It should be noted that in spite of its limits towards old TEs, dnaPipeTE is well-suited for this analysis as it is not biased by highly repetitive new TE families that are challenging to assemble. In the revised manuscript, we now emphasize the limitations of dnaPipeTE and discuss the consequences on our results. See lines 359-374 (reported above) and lines 449-455: “On the other hand, it is conceivable the avian TE diversity to be underappreciated due to the limits of sequencing technologies used so far in resolving complex repeat-rich regions. For instance, employment of long-reads technologies allowed to reveal more extended repeated regions that were previously ignored with short read assemblies (Kapusta and Suh 2017; Benham et al. 2024). Besides, quite large fractions might indeed be satellite sequences constituting relevant fractions of the genome that are challenging to identify with reference- or read-based methods (Edwards et al. 2025).”

Finally, in a preliminary work on the dipteran species, we showed that the TE content estimated with dnaPipeTE is generally similar to that estimated from the assembly with earlGrey (Baril et al., 2024 Mol Biol Evol 38: msae068) across a good range of genome sizes going from drosophilid-like to mosquito-like (TE genomic percentage: Pearson’s r = 0.88, p-value = 1.951e-10; TE base pairs: Pearson’s r = 0.90, p-value = 3.573e-11; see also the corrected Supplementary Figure S2 and new Supplementary Figure S3). While TEs for these species are probably dominated by recent to moderately recent TEs, Ae. albopictus is an outlier for its genome size and the estimations with the two methods are largely consistent. However, the computation time required to estimate TE content using EarlGrey was significantly longer, with a ~300% increase in computation time, making it a very costly option (a similar issue applicable to other assembly-based annotation pipelines). Given the rationale presented above, we decided to use dnaPipeTE instead of EarlGrey.

**Recommendations for the authors:**

**Reviewer #1 (Recommendations for the authors):**
Since I am not an expert in the field, some of these comments may simply reflect a lack of understanding on my part. However, in those cases, I hope they can help the authors clarify important points. I did have a bunch of comments concerning the complexity of the relationship between TEs and their hosts that would likely affect TE content, but I ended up deleting most of them because they were covered in the discussion. However, I do think that in setting up the paper, particularly given the results, it might have been useful to introduce those issues in the introduction. That is to say, treating TEs as a generic mutagen that will fit into a relatively simple model is unlikely to be correct. What will ultimately be more interesting are the particulars of the ways that the relationships between TEs and their host evolve over time. Finally, given the huge variation in plant genes with respect to genome size and TE content, along with really interesting variation in deletion rates, I'm surprised that they were not included. I get that you have to draw a line somewhere, and this work builds on a bunch of other work in animals, but it seems like a missed opportunity.

We chose to restrict the introduction to the rationale behind the MHH as it is the starting point and focus of the manuscript. Because the aspects of the complexity of TE-host relationships are only covered in a speculative way, we limited them to the discussion but it is true that introducing them at the very beginning gives a more comprehensive overview. The introduction now includes a few sentences about lineage-specific selective effect of TEs and TE-host evolution (lines 83-86): “On top of that, an alternative TE-host-oriented perspective is that the accumulation of TEs in particular depends on their type of activity and dynamics, as well as on the lineage-specific silencing mechanisms evolved by host genomes (Ågren and Wright 2011).”

Page 4. "The MHH is highly popular..." Evidence for this? It is fine as is, but it could also be seen as a straw man argument. Perhaps make clear this is an opinion of the authors?

That MHH is popular and well-known is more a fact than an opinion: the original paper by Lynch and Conery (2003) and “The origins of genome architecture” by Lynch (2007) have respectively 1872 and 1901 citations to the present date (04/03/2025). Besides, the MHH is often invoked in highly cited reviews about TEs, e.g. Bourque et al., 2018 Genome Biol 19:1-12; Wells and Feschotte, 2020 Annu Rev Genet 54: 539-561.

Page 4. "on phylogenetically very diverse datasets..." Given the fact that even closely related plants can show huge variation in genome size, it's a shame that they weren't included here. There are also numerous examples of closely related plants that are obligate selfers and out-crossers.

This is true, and some studies already tested MHH in specific plant groups (Ågren et al., 2014 BMC Genom 15: 1-9; Hu et al., 2011 Nat Genet 43: 476-481; Wright et al., 2008 Int J Plant Sci 169: 105-118), including selfers vs out-crossers cases (Glémin et al., 2019 Evolutionary genomics: statistical and computational methods: 331-369). Further development in this kingdom would be interesting. However, the boundary was set to metazoans since the very beginning of analyses to maintain a large phylogenetic span and a manageable computational burden. Furthermore, some of the included animal clades are supposed to display good Ne contrasts according to known LHTs or to previous literature: for instance, the very different Ne of mammals and insects, as well as more narrowed examples like Drosophilidae and solitary vs eusocial hymenopterans.

Page 6. "species-poor, deep-branching taxa were excluded" I see why this was done, as these taxa would not provide close as well as distant comparisons, but I would have thought they might have provided some interesting outlying data. As the geneticists say, value the exceptions.

The reason to exclude them was not only that they would solely provide very distant comparisons. The lack of a rich and balanced sampling would imply calculating nucleotide substitution rates over hundreds of millions of years, which typically lead to saturation of synonymous sites. In case of saturation of synonymous sites, the synonymous divergence will be underestimated, and therefore, the dN/dS ratio no longer a valuable estimate of N_e_. Outside vertebrates and insects, the available genomes in a clade would mostly correspond to a few species from an entire phylum, making it challenging to estimate dN/dS and to correlate present day genome size with Ne estimated over hundreds of millions of years.

Figure 1. What are the scaling units for each of these values? I get that dN/dS is between 0 and 1, but what about genome sizes? Are these relative sizes? Are TE content values a percent of the total? This may be mentioned elsewhere, but I think it is worth putting that information here as well.

Thanks for pointing this out. Both genome sizes and TE contents are in bp, we added this information in the legend of the figure.

Page 8. TE content estimates are invariably wrong given the diversity of TEs and, in many genomes, the presence of large numbers of low copy number "dead" elements. If that varies between taxa, this could cause problems. Given that, I would have liked to see the protocols used here be compared to a set of "gold standard" genomes with exceptionally well-annotated TEs (Humans and *D. melanogaster*, for instance).

As already mentioned, dnaPipeTE is indeed biased towards young TEs (elements older than 25-30% are generally not detected). TE content can therefore be underestimated, especially in those genomes that tend to accumulate TEs rather than getting rid of them. Although most of them do not have “gold-standard” genomes, a comparison of dnaPipeTE with TE annotations from assemblies is already provided for a subset of species. Some variation can be present - see Supplemental Figure S6 and comments of Reviewer#3 about detection of satellite sequences. However, the subset covers a good range of genome sizes and overall dnaPipeTE emerges as an appropriate tool to characterize the general patterns of repeat content variation.

Page 11. "close to 1 accounts for more..." I would say "closer" rather than "close".

Agreed and changed.

Page 11. "We therefore employed this parameter..." I know you made the point earlier, but maybe reiterate the general point here that selection is lower on average with a lower effective population size. Actually, I'm wondering if we don't need a different term for long-term net effective population size, which dN/dS is measuring.

We reiterated here the relationship among dN/dS, Ne and magnitude of selection (lines 200-204): “a dN/dS closer to 1 accounts for more frequent accumulation of mildly deleterious mutations over time due to increased genetic drift, while a dN/dS close to zero is associated with a stronger effect of purifying selection. We therefore employed this parameter as a genomic indicator of N_e_, as the two are expected to scale negatively between each other.”

Page 11. "We estimated dN/dS with a mapping method..." I very much appreciate that the authors are using the same pipeline for the analysis of all of these taxa, but I would also be interested in how these dN/dS values compare with previously obtained values for a subset of intensively studied taxa.

The original publication of the method demonstrated that dN/dS estimations using mapping are highly similar to those obtained with maximum likelihood methods, such as implemented in CODEML (Romiguier et al., 2014 J Evol Biol 27: 593-603). Below is the comparison for 16 vertebrate species from Figuet et al. (2016 Mol Biol Evol 33: 1517-1527), where dN/dS are reasonably correlated (slope = 0.57, adjusted-R^2^ = 0.39, p-value=0.006). That being said, some noise can be present as the compared genes and the phylogeny used are different. Although we expect some value between 0 and 1, some range of variation is to be expected depending on both the species used and the markers, as substitution rates and/or selection strength might be different. Differences in dN/dS for the same species would not necessarily imply an issue with one of the methods.

Page 12. " As expected, Bio++ dN/dS scales positively with..." Should this be explicitly referenced earlier? I do see that references mentioning both body mass and longevity are included earlier, but the terms themselves are not.

We added a list of the expected correlations for dN/dS and LHTs at the beginning of the paragraph (lines 205-208): “In general, dN/dS is expected to scale positively with body length, age at first birth, maximum longevity, age at sexual maturity and mass, and to scale negatively with metabolic rate, population density and depth range.”

Page 12. "dN/dS estimation on the trimmed phylogeny deprived of short and long branches results in a stronger correlation with LHTs, suggesting that short branches..." and what about the long branches? Trimming them helps because LHTs change over long periods of time?

Trimming of long branches should avoid saturation in the signal of synonymous substitutions if present (whereby increase in dN is not parallelled by corresponding increase in dS due to depletion of all sites). Excluding very long branches was one of the reasons why we excluded taxonomic groups with few species. See lines 131-133: “For reliable estimation of substitution rates, this dataset was further downsized to 807 representative genomes as species-poor, deep-branching taxa were excluded”. Correlating present-day genome size with Ne estimates over long periods of time could weaken a potential correlation. However, exploratory analyses (not included) did not indicate that excluding long branches improved the relationship between Ne and genome size/TE content. The rationale is explained in Materials and Methods but was wrongly formulated. We rephrased it and added a reference (lines 636-638): “Estimation of dN/dS on either very long or short terminal branches might lead to loss of accuracy due to branch saturation (Weber et al. 2014) or to a higher variance of substitution rates, respectively”.

Table 2. "Expected significant correlations are marked in bold black; significant correlations opposite to the expected trend are marked in bold red." Expected based on the initial hypothesis? Perhaps frame it as a test of the hypothesis?

As per the comment above, we added a sentence in the main text to clarify the expected correlations for dN/dS and LHTs (lines 205-208): “In general, dN/dS is expected to scale positively with body length, age at first birth, maximum longevity, age at sexual maturity and mass, and to scale negatively with metabolic rate, population density and depth range.”. The second expected correlation is that between dN/dS and genome size/TE content, which is stated at the beginning of paragraph 2.5 (lines 244-245): “If increased genetic drift leads to TE expansions, a positive relationship between dN/dS and TE content, and more broadly with genome size, should be observed.”.

Page 14. "Based on the available traits, the two kinds of Ne proxies analyzed here correspond in general..." the two kinds being dN/dS and a selection of LHT?

We rephrased the sentence as such (lines 233-234): “Based on the available traits, the estimations of dN/dS ratios obtained using two different methods correspond in general to each other”.

Table 3. Did you explain why there is a distinction between GC3-poor and GC3-rich gene sets?

No, the explanation is missing, thank you for pointing it out. The choice comes from the observations made by Mérel et al. (2024 biorXiv: 2024-01), who do find a stronger relationship between dN/dS and genome size in *Drosophila* using the same tool (Coevol) in GC3-poor genes than in GC3-rich ones or in random sets of genes exhibiting heterogeneity in GC3 content. There are several possible explanations for this. First, mixing genes with various base compositions in the same concatenate can alter the calculation of codon frequency and impair the accuracy of the model estimating substitution rates.

Moreover, base composition and evolutionary rates may not be two independent molecular traits, at the very least in Drosophila, and more generally in species experiencing selection on codon bias. Because optimal codons are enriched in G/C bases at the third position (Duret and Mouchiroud, 1999 PNAS 96: 4482-4487), GC3-rich genes are likely to be more expressed and therefore evolve under stronger purifying selection than GC3-poor genes in *Drosophila*.

Accordingly, Merel and colleagues observed significantly higher dN/dS estimates for GC3-poor genes than for GC3-rich genes. Additionally, selection on codon usage acting on these highly expressed genes, that are GC3-rich, violates the assumed neutrality of dS. This implies that dN/dS estimates based on genes under selection on codon bias are likely less appropriate proxies of Ne than expected.

Although some of these observations may be specific to *Drosophila*, this criterion was taken into consideration as taking restricted gene subsets was required for Coevol runs. We added this explanation in materials and methods (lines 723-738).

Page 16. "Coevol dN/dS scales negatively with genome size across the whole dataset (Slope = -0.287, adjusted-R^2^ = 0.004, p-value = 0.039) and within insects" Should I assume that none of the other groups scale negatively on their own, but cumulatively, all of them do?

Yes, and this is an “insect-effect”: the regression of the whole dataset is negative but it is not anymore when insects are removed (with the model still being far from significant).

Page 16. "Overall, we find no evidence for a recursive association of dN/dS with genome size and TE content across the analysed animal taxa as an effect of long-term Ne variation." I get the point, but this is starting to feel a bit circular. What you see is a lack of an association between dN/dS and TE content, but what do you mean by "as an effect of..." here? You are using dN/dS as a proxy, so the wording here feels odd.

See the reply below.

Page 17. I'm not sure that "effect" here is the word to use. You are looking at associations, not cause-effect relationships. Certainly, dN/dS is not causing anything; it is an effect of variation in purifying selection.

Agreed, dN/dS is the ratio reflecting the level of purifying selection, not the cause itself. dN/dS is employed here as the independent variable in the correlation with genome size or TE content. dN/dS has an “effect” on the dependent variables in the sense that it can predict their variation, not in the sense that it is causing genome size to vary. We rephrased this and similar sentences to avoid misunderstandings (changes are highlighted in the revised text).

Page 17. "Instead, mammalian TE content correlates positively with metabolic rate and population density, and negatively with body length, mass, sexual maturity, age at first birth and longevity." I guess I'm getting tripped up by measures of current LHTs and historical LHTs which, I'm assuming, varies considerably over the long periods of time that impact TE content evolution.

PIC analyses can be considered as correlations on current LHTs as we compare values (or better, contrasts) at the tips of phylogenies. In the case of Coevol, traits are inferred at internal nodes, in such a way that the model should take into account the historical variation of LHTs, too.

Page 18. "positive effect of dN/dS on recent TE insertions..." Again, this is not a measure of the effect of dN/dS on TE insertions, it is a measure of correlation. I know it's shorthand, but in this case, I think it really matters that we avoid making cause inferences.

We have rephrased this as ”...very weak positive correlation of dN/dS with recent TE insertions…”.

Page 18. "are consistent with the scenarios depicted by genome size and overall TE content in the corresponding clades." Maybe be more explicit here at the very end of the results about what those scenarios are.

Correlating the recent TE content with dN/dS and LHTs basically recapitulates the relationship found using the other genomic traits (genome size and overall TE content). We have rephrased the closing sentence as “Therefore, the coevolution patterns between population size and recent TE content are consistent with the pictures emerging from the comparison of population size proxies with genome size and overall TE content in the corresponding clades” (lines 312-315).

Page 19. "However, the difficulty in assembling repetitive regions..." I would say the same is true of TE content, which is almost always underestimated for the same reasons.

“Repetitive regions” is here intended as an umbrella term including all kinds of repeats, from simple ones to transposable elements.

Page 20. "repeat content has a lower capacity to explain size compared to other clades." Perhaps, but I'm not convinced this is not due to large numbers of low copy number elements, perhaps purged at varying rates. Are we certain that dnaPipeTE would detect these? Have rates of deletion in the various taxa examined been estimated?

It is possible that low copy number elements are detected differently, according to the rate of decay in different species and depending also on the annotation method (indeed low copy families are less likely to be captured during read sampling by dnaPipeTE). A negative correlation between assembly size and deletion rate was observed in birds (Ji et al., 2023 Sci Adv 8: eabo0099). So we should expect a rate of TE removal inversely proportional to genome size, a positive correlation between TE content and genome size, and negative relationship between TE content and deletion rate, too. The relationship of TE content with deletion rate and genome size however appears more complex than this, even this paper using assembly-based TE annotations. However, misestimations of repeat content are also potentially due to the limited capacity of dnaPipeTE of detecting simple and low complexity repeats (see comments from Reviewer#3), which might be important genomic components in birds (see a few comments below).

Page 21. "DNA gain, and their evolutionary dynamics appear of prime importance in driving genome size variation." How about DNA loss over time?

See response to the comment below.

Page 22. "in the latter case, the pace of sequence erosion could be in the long run independent of drift and lead to different trends of TE retention and degradation in different lineages." Ah, I see my earlier question is addressed here. How about deletion as a driver as well?

Deletion was not investigated here. However, deletion processes are surely very different across animals and their impact merits to be studied as well within a comparative framework. Small scale deletion events have even been proposed to contrast the increase in genome size by TE expansion (Petrov et al., 2002 Theor Popul Biol 61: 531-544). In fact, their magnitude would not be high enough to effectively contrast processes of genome expansion in most organisms (Gregory, 2004 Gene 324: 15-34). However, larger-scale deletions might play an important role in genome size determinism by counterbalancing DNA gain (Kapusta et al., 2017 PNAS 114: E1460-E1469; Ji et al., 2023 Sci Adv 8: eabo0099). For sake of space we do not delve in detail into this issue, but we do provide some perspectives about the role of deletion (see lines 518-521 and 535-541).

Page 22. "however not surprising given the higher variation of TE load compared to the restricted genome size range." I admit, I'm struggling with this. If it isn't genes, and it isn't satellites, and it isn't TEs, what is it?

Most birds having ~1Gb genomes and displaying very low TE contents. Other studies annotated TEs in avian genome assemblies and also found a not so strong correlation between amount of TEs and genome size (Ji et al., 2023 Sci Adv 8: eabo0099, Kapusta and Suh, 2016 Ann N Y Acad Sci 1389: 164-185). It is possible that the TE diversity is underappreciated in birds due to the limits of sequencing technologies used so far in resolving complex repeat-rich regions. For instance, employment of long-reads technologies allowed to reveal more extended repeated regions that were previously ignored with short read assemblies (Kapusta and Suh, 2016 Ann N Y Acad Sci 1389: 164-185). Besides, quite large fractions might indeed be satellite sequences constituting relevant fractions of the genome (Edwards et al., 2025 biorXiv: 2025-02). We added this perspective in the discussion (lines 446-455): “As previous studies find relatively weak correlations between TE content and genome size in birds (Ji et al. 2022; Kapusta and Suh 2017), it is possible for the very narrow variation of the avian genome sizes to impair the detection of consistent signals. On the other hand, it is conceivable the avian TE diversity to be underappreciated due to the limits of sequencing technologies used so far in resolving complex repeat-rich regions. For instance, employment of long-reads technologies allowed to reveal more extended repeated regions that were previously ignored with short read assemblies (Kapusta and Suh 2017; Benham et al. 2024). Besides, quite large fractions might indeed be satellite sequences constituting relevant fractions of the genome that are challenging to identify with reference- or read-based methods (Edwards et al. 2025).” See also responses to Reviewer#3’s concerns about dnaPipeTE.

Page 24. "Our findings do not support the quantity of non-coding DNA being driven in..." Many TEs carry genes and are "coding".

Yes. Non-coding DNA intended as the non-coding portion of genomes not directly involved in organisms’ functions and fitness (in other words sequences not undergoing purifying selection). TEs do have coding parts but are in most part molecular parasites hijacking hosts’ machinery.

Page 25. "There is some evidence of selection acting against TEs proliferation." Given that the vast majority of TEs are recognized and epigenetically silenced in most genomes, I'd say the evidence is overwhelming. Here I suspect you mean evidence for success in preventing proliferation. Actually, since we know that systems of TE silencing have a cost, it might be worth considering how the costs and benefits of these systems may have influenced overall TE content.

We meant selection against TE proliferation in the making, notably visible at the level of genome-wide signatures for relaxed/effective selection. We rephrased it as “Evidence for signatures of negative selection against TE proliferation exist at various degrees.” (line 543).

**Reviewer #3 (Recommendations for the authors):**
Page 14: Please define GC3-rich and GC3-poor gene sets and how they were established, as well as why the analyses were conducted separately on GC3-rich and GC3-poor genes.

We added a detailed explanation for the choice of GC3-rich and GC3-poor genes (see modified section Methods - Phylogenetic independent contrasts and Coevol reconstruction, lines 723-738).

“Genes were selected according to their GC content at the third codon position (GC3). Indeed, mixing genes with heterogeneous base composition in the same concatenate might result in an alteration of the calculation of codon frequencies, and consequently impair the accuracy of the model estimating substitution rates (Mérel et al. 2024). Moreover, genes with different GC3 levels can reflect different selective pressures, as highly expressed genes should be enriched in optimal codons as a consequence of selection on codon usage. In *Drosophila*, where codon usage bias is at play, most optimal codons present G/C bases at the third position (Duret and Mouchiroud, 1999), meaning that genes with high GC3 content should evolve under stronger purifying selection than GC3-poor genes. Accordingly, Mérel et al. (2024) do find a stronger relationship between dN/dS and genome size when using GC3-poor genes, as compared to GC3-rich genes or gene concatenates of random GC3 composition. Finally, dN/dS can be influenced by GC-biased gene conversion (Bolívar et al. 2019; Ratnakumar et al. 2010), and the strength at which such substitution bias acts can be reflected by base composition. For these reasons, two sets of 50 genes with similar GC3 content were defined in order to employ genes undergoing similar evolutionary regimes.”

Please add lines between columns and rows in tables. Table 3 is especially difficult to follow due to its size, and lines separating columns and rows would vastly help with readability.

We added lines delimiting cells in all the main tables.

Throughout the text and figures, please be consistent with either scientific names or common names for lineages or clades.

Out of the five groups, for four of them the common name is the same as the scientific one (except Aves/birds).

Regarding the title for section 3.1, I don't believe "underrate" is the best word here. I find this title confusing.

We replaced the term “underrate” with “underestimate” in the title.

The authors report that read type (short vs. long) does not have a significant effect on assembly size relative to C-value. However, the authors (albeit admittedly in the discussion) removed lower-quality assemblies using a minimum N50 cutoff. Thus, this lack of read-type effect could be quite misleading. I strongly recommend the authors either remove this analysis entirely from the manuscript or report results both with and without their minimum N50 cutoff. I expect that read type should have a strong effect on assembly size relative to C-value, especially in mammals where TEs and satellites comprise ~50% of the genome.

Yes, it's likely that if we took any short-read assembly, we would have a short-read effect. We do not mean to suggest that in general short reads produce the same assembly quality as long reads, but that in this dataset we do not need to account for the read effect in the model to predict C-values. Adding the same test including all assemblies will be very time-consuming because C-values should be manually checked as already done for the species. If we removed this test, readers might wonder whether our genome size predictions are not distorted by a short-read effect. We now make it clear that this quality filter likely has an outcome on our observations: “This suggests that the assemblies selected for our dataset can mostly provide a reliable measurement of genome size, and thus a quasi-exhaustive view of the genome architecture.” (lines 333-335).

There seem to be some confusing inconsistencies between Supplementary Table S2 and Supplementary Figure S2. In Supplementary Table S2, the authors report ~24% of the *Drosophila* pectinifera genome as unknown repeats. This is not consistent with the stacked bar plot for D. pectinifera in Supplementary Figure S2.

True, the figure is wrong, thank you for spotting the error. The plot of Supplemental Figure S2 was remade with the correct repeat proportions as in Supplementary Tables S2 and S4. Because the reference genome sizes on which TE proportions are calculated are different for the two methods, we added another supplemental figure showing the same comparison in Kbp (now Supplemental Figure S3).

At the bottom of page 20: "many species with a high duplication score in our dataset correspond to documented duplication" How many?

Salmoniformes (9), Acipenseriformes (1), Cypriniformes (3) out of 23 species with high duplication score. It’s detailed in the results (lines 193-196): “Of the 24 species with more than 30% of duplicated BUSCO genes, 13 include sturgeon, salmonids and cyprinids, known to have undergone whole genome duplication (Du et al. 2020; Li and Guo 2020; Lien et al. 2016), and five are dipteran species, where gene duplications are common (Ruzzante et al. 2019).”

Top of page 21: "However, the contribution of duplicated genes to genome size is minimal compared to the one of TEs, and removing species with high duplication scores does not affect our results: this implies that duplication does not impact genome size strongly enough to explain the lack of correlation with dN/dS." This sentence is confusing and needs rewording.

We reworded the sentence (lines 383-384): “this implies that duplication is unlikely to be the factor causing the relationship between genome size and dN/dS to deviate from the pattern expected from the MHH”.

Beginning of section 3.3: "Our dN/dS calculation included several filtering steps by branch length and topology: indeed, selecting markers by such criteria appears to be an essential step to reconcile estimations with different methodologies" A personal communication is cited here. Are there really no peer-reviewed sources supporting this claim?

This mainly comes from a comparison of dN/dS calculation with different methods (notably ML method of bpp vs Coevol bayesian framework) on a set of Zoonomia species. We observed that estimations with different methods appeared correlated but with some noise: filtering out genes with deviant topologies (by a combination of PhylteR and of an unpublished Bayesian shrinkage model) reconciled even more the estimations obtained from different methods. Results are not shown here but the description of an analogous procedure is presented in Bastian, M. (2024). Génomique des populations intégrative: de la phylogénie à la génétique des populations (Doctoral dissertation, Université lyon 1) that we added to the references.

Figure 2 needs to be cropped to remove the vertical gray line on the right of the figure as well as the portion of visible (partly cropped) text at the top. What is the "Tree scale" in Figure 1?

Quality of figure 2 in the main text was adjusted. The tree scale is in amino acid substitutions, we added it in the legend of the figure.

It is also unclear whether the authors used TE content or overall repeat content for their analyses.

The overall repeat content includes both TEs and other kinds of repeats (simple repeats, low complexity repeats, satellites). The contribution of such other repeats to the total content is generally quite low for most species compared to that of TEs (only 13 genomes in all dataset have more than 3% of “Other” repeats). Conversely, the “other” repeats were not included in the recent content since the divergence of a copy from its consensus sequence is pertinent only for TEs.